# Is Your Diffusion Sampler Actually Correct?
# A Sampler-Centric Evaluation of Discrete Diffusion Language Models

**Luhan Tang** [1]   **Longxuan Yu** [1]   **Shaorong Zhang** [1]   **Greg Ver Steeg** [1]

## Abstract

Discrete diffusion language models (dLLMs) provide a fast and flexible alternative to autoregressive models (ARMs) via iterative denoising with parallel updates. However, their evaluation is challenging: existing metrics conflate denoiser approximation error with sampler-induced error from the sampling dynamics, a problem that does not arise for ARMs whose autoregressive sampling exactly reflects the learned probability model. We introduce a sampler-centric oracle framework that replaces learned denoisers with an exact Hidden Markov Model posterior derived from a ground-truth Markov chain, isolating sampler-induced error in a controlled setting. We show that few-step discrete diffusion samplers are not distributionally correct even under an oracle denoiser, with transition-level mismatch that vanishes only as the number of steps approaches the sequence length. Moreover, improvements in negative log-likelihood (NLL), generative perplexity (GenPPL), or MAUVE do not imply correct sampling. Code is available at https://luhantang.github.io/dllm_sampler/.

## 1. Introduction

Discrete diffusion language models (dLLMs) have emerged as an alternative to autoregressive models (ARMs) (Radford et al., 2019; Grattafiori et al., 2024; Team et al., 2024) for discrete sequence generation. Rather than strict left-to-right factorization, dLLMs generate sequences through iterative denoising, enabling parallel updates across multiple positions (Austin et al., 2021; Campbell et al., 2022; Chang et al., 2022; Sahoo et al., 2024; Arriola et al., 2025). This formulation offers potential advantages in inference efficiency and supports capabilities that are difficult to realize with ARMs, including bidirectional context utilization, iterative revision, and global constraint satisfaction (Ghazvininejad et al., 2019; Nie et al., 2026). Recent dLLMs have demonstrated competitive performance on standard language modeling benchmarks, motivating further investigation into their properties and limitations.

Despite these architectural differences, existing dLLMs are predominantly evaluated using protocols developed for ARMs, assessing the properties of the final generated samples through task-based metrics such as zero-shot accuracy (Ye et al., 2025; Nie et al., 2026), likelihood-based metrics such as perplexity (Austin et al., 2021; Lou et al., 2023; Sahoo et al., 2024), generation-based metrics such as generative perplexity (GenPPL) or MAUVE (Pillutla et al., 2021; Lou et al., 2023; Sahoo et al., 2024; Wang et al., 2026), and surface-level metrics such as BLEU or $n$-gram diversity (Li et al., 2022). Underlying all of these is a shared assumption: that observed generation error faithfully reflects model quality. This assumption is justified for ARMs, where exact ancestral sampling introduces no additional error — the training objective directly optimizes the model distribution, and sampler correctness follows by construction.

dLLMs, however, violate this assumption by design. Their sampling procedures involve approximations: parallel decoding assumes conditional independence among masked positions, and finite-step generation discretizes continuous-time dynamics. Such approximations introduce systematic bias independent of model quality, obscuring the contribution of the sampler to generation error.

As a result, standard metrics conflate two distinct sources of error that are separable only for dLLMs:

- **Model approximation error:** the learned denoiser $p_\theta(x_0 \mid z_t)$ deviates from the true posterior $p_0(x_0 \mid z_t)$ due to limited capacity, optimization constraints, or insufficient training.

- **Sampling Dynamics Error:** introduced by the generation procedure itself, such as conditional independence assumptions in parallel decoding or discretization errors from finite time steps.

[1]University of California, Riverside. Correspondence to: Greg Ver Steeg <gregoryv@ucr.edu>.

*Proceedings of the 43rd International Conference on Machine Learning*, Seoul, South Korea. PMLR 306, 2026. Copyright 2026 by the author(s).

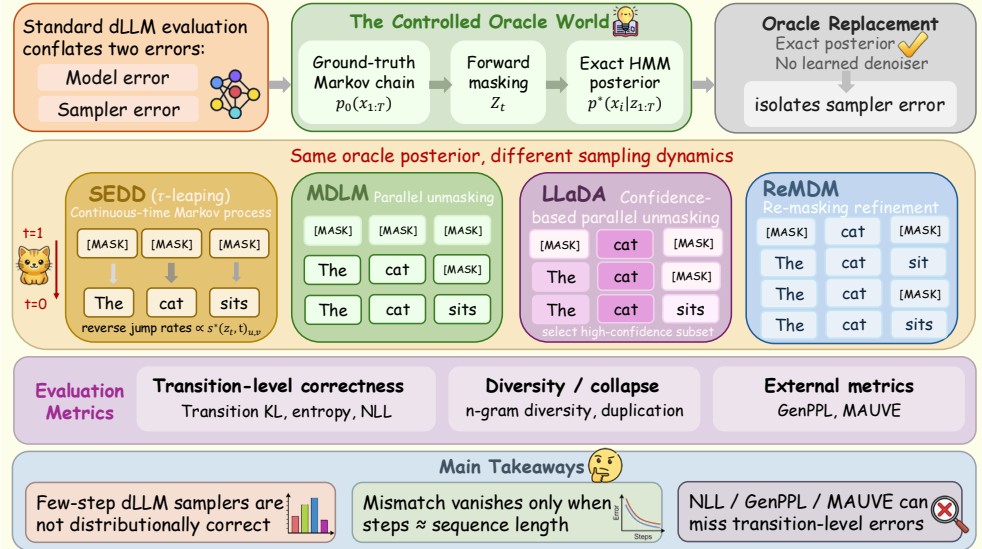

*Figure 1.* **Overview of our sampler-centric oracle evaluation framework for dLLMs.** We construct a controlled oracle world using a ground-truth Markov chain and exact HMM posterior inference to isolate sampler-induced error. Different samplers (SEDD, MDLM, LLaDA, and ReMDM) are then evaluated under the same oracle posterior using transition-level correctness, diversity, and external metrics.

Yet existing metrics, observing only final outputs, offer no means to attribute generation error to either source, leaving the sampler's contribution systematically unexamined.

To address this, we propose an **oracle evaluation framework** that eliminates model approximation error by construction. We define the data distribution as a discrete Markov chain, for which exact posteriors under partial observations can be computed using a Hidden Markov Model (HMM). Replacing learned denoisers with this oracle isolates sampling dynamics as the sole source of distributional deviation, enabling controlled comparison across sampler designs.

Our contributions are as follows:

1. We introduce a **sampler-centric oracle evaluation framework** for dLLMs, which replaces learned denoisers with an exact HMM posterior from a ground-truth Markov chain, isolating sampler-induced error and enabling controlled comparison across sampler designs under method-consistent settings.

2. We show that **few-step discrete diffusion is not distributionally correct even under an exact oracle denoiser**. Across SEDD, MDLM, LLaDA, and ReMDM, substantial transition-level mismatch persists at small step counts and vanishes only as the number of diffusion steps approaches the sequence length.

3. We show that **existing evaluation metrics do not imply correct sampling**: fluent samples can exhibit substantial transition-level mismatch. Likelihood-based metrics such as NLL and generation-based met-

rics such as GenPPL can both improve under biased sampling, while MAUVE may remain insensitive to transition-level errors.

## 2. Background and Problem Setup

### 2.1. Diffusion Language Models

We focus on dLLMs for discrete sequence generation over a finite vocabulary. Although these models differ in their reverse-time denoiser parameterization and sampling mechanisms, SEDD (Lou et al., 2023), MDLM (Sahoo et al., 2024), LLaDA (Nie et al., 2026), and ReMDM (Wang et al., 2026) share a common two-stage forward–reverse generation framework.

Throughout the paper, we use $t$ to index diffusion steps, $u \in \{1, \ldots, T\}$ to index token positions within a sequence, and $v \in \mathcal{V}$ to denote vocabulary elements (token values).

**Forward process.** The forward process progressively corrupts a clean sequence $x_0$ into a noisy state $z_t$. It is typically modeled as a factorized Markov process where tokens are independently corrupted (e.g., replaced by a mask token):

$$q(z_t \mid x_0) = \prod_{u=1}^{T} q(z_t^{(u)} \mid x_0^{(u)}), \qquad (1)$$

where the marginal probability of a token remaining clean decays according to a schedule $\alpha_t$. This process is assumed to be fixed and is used solely for training the denoiser.

**Backward process and sampling.** Generation is performed by reversing this corruption. The true posterior $p(x_0 \mid z_t)$ is

intractable, so a denoiser $p_\theta$ is trained to approximate it:

$$p_\theta(x_0 \mid z_t) \approx p(x_0 \mid z_t). \tag{2}$$

The sampler then uses this prediction to update the sequence state $z_t \to z_{t-1}$. The specific mechanisms differ across methods:

**(1) SEDD** models the backward process as a continuous-time Markov process. It parameterizes the transition rates via a concrete score function representing probability ratios:

$$s_\theta(z_t, t)_{u,v} \approx \frac{p_t(z_t^{u \leftarrow v})}{p_t(z_t)}, \tag{3}$$

where $z_t^{u \leftarrow v}$ denotes the sequence obtained from $z_t$ by replacing its $u$-th token with $v \in \mathcal{V}$. The sampler then uses these scores to drive stochastic jumps from masks to tokens via $\tau$-leaping.

**(2) MDLM** formulates discrete generation via masked diffusion. Given a masked sequence $z_t$, the denoiser predicts a clean-token distribution at every position:

$$p_\theta(x_0 \mid z_t) = \prod_{u=1}^{T} p_\theta(x_0^{(u)} \mid z_t), \tag{4}$$

where each factor is a categorical distribution over vocabulary elements. During training and sampling, the effective updates are applied to the masked position set $\mathcal{M}_t = \{u : z_t^{(u)} = [\text{M}]\}$, while unmasked positions are carried over. Here, $[\text{M}] \in \mathcal{V}$ denotes the mask token value.

**(3) LLaDA** follows the masked-diffusion denoising form in Equation (4), where a mask predictor predicts masked tokens in parallel given $z_t$. Unlike MDLM, its sampler uses confidence-based selection: high-confidence predictions are accepted, while low-confidence positions remain masked for later steps.

**(4) ReMDM** modifies the sampler by allowing already decoded tokens to be masked again. In particular, when $z_t^{(u)} \neq [\text{M}]$, the reverse step can be written as

$$z_s^{(u)} \sim (1 - \sigma_t)\, p_\theta(x_0^{(u)} \mid z_t) + \sigma_t\, \delta_{[\text{M}]}. \tag{5}$$

Thus, $\sigma_t$ controls the probability of remasking a decoded token, allowing the sampler to undo previous decisions and perform iterative refinement. For masked positions, ReMDM uses the corresponding schedule-dependent posterior involving $\alpha_t$ and $\alpha_s$.

## 2.2. Role of the Sampler

It is important to distinguish the denoiser from the sampler. The denoiser provides local conditional predictions, such as $p_\theta(x_0 \mid z_t)$, while the sampler defines how these predictions are used to construct the reverse update path. Thus, errors

can arise not only from an inaccurate denoiser, but also from the sampling procedure itself. We refer to the latter as *sampling error*, and next introduce a controlled framework for isolating it.

# 3. Method: A Controlled Framework to Isolate Sampler Error

We define a ground-truth Markov-chain data distribution and compute exact posterior token marginals under masking using HMM forward–backward inference. These marginals serve as oracle denoisers, allowing us to evaluate sampler error without learned-model approximation.

## 3.1. Ground-Truth Markov Chain Prior

**Ground-truth Markov prior.** We consider a ground-truth data distribution $p_0$ over sequences $x_{1:T} = (x_1, \ldots, x_T) \in \mathcal{V}^T$, defined as a discrete-time Markov chain:

$$p_0(x_{1:T}) = \pi_0(x_1) \prod_{i=2}^{T} P(x_i \mid x_{i-1}), \tag{6}$$

where $\mathcal{V}$ is a finite vocabulary, $\pi_0$ denotes the initial distribution, and $P$ the transition kernel.

**Computational considerations.** To balance expressiveness, ergodicity, and computational tractability (Norris, 1998), we specify a concrete instantiation of the transition kernel by constructing a sparse Markov prior. We define an effective transition kernel $P'$ that retains only the top-$K$ outgoing transitions per state, with $K \ll V$, and includes a small teleport component $\varepsilon$ (Page et al., 1999):

$$P'(v \mid v') = (1 - \varepsilon)\, P_{\text{top-}K}(v \mid v') + \varepsilon\, \nu(v), \qquad \varepsilon \in (0, 1). \tag{7}$$

This construction reduces the cost of sampling and oracle evaluation from $\mathcal{O}(V^2)$ to $\mathcal{O}(V)$ per time step.

Throughout the paper, $P'$ defines the ground-truth data distribution $p_0$. Thus, the top-$K$ truncation and teleport mixing in Eq. (7) are part of the oracle construction, not inference approximations.

## 3.2. HMM Posterior Under Partial Observations

The forward masking process applied to a Markov chain induces a Hidden Markov Model (HMM) (Eddy, 1996; Bishop & Nasrabadi, 2006; Murphy, 2012). The clean sequence forms the latent states, while masked tokens correspond to missing observations. Under this formulation, the exact posterior under partial observations can be computed via standard forward–backward smoothing (Rabiner, 1989).

In this subsection, we fix a diffusion step $t$ and view the resulting noisy sequence $z_t$ as a partially observed sequence

$z_{1:T}$ for HMM inference. For notational convenience within this HMM formulation, we use $i$ to index sequence positions, instead of $u$ used elsewhere in the paper.

**Model definition.** We treat the ground-truth Markov chain in Equation (6) as the latent state sequence of an HMM. Let $x_{1:T} \in \mathcal{V}^T$ denote the latent clean sequence, and let $z_{1:T} \in (\mathcal{V} \cup \{\texttt{MASK}\})^T$ denote the partially observed sequence, where $z_t = \texttt{MASK}$ indicates that the clean token $x_t$ is unobserved. Observed positions satisfy $z_t = x_t$.

Define the set of revealed positions

$$\mathcal{R}(z) = \{i \in \{1, \dots, T\} : z_i \neq \texttt{MASK}\}, \tag{8}$$

and the corresponding hard-evidence event

$$\mathcal{E}(z) = \{x_{1:T} \in \mathcal{V}^T : x_i = z_i \text{ for all } i \in \mathcal{R}(z)\}. \tag{9}$$

Under this hard evidence, the posterior distribution over latent state sequences is

$$p(x_{1:T} \mid \mathcal{E}(z)) \propto \pi_0(x_1) \prod_{i=2}^{T} P'(x_i \mid x_{i-1}) \prod_{i=1}^{T} \phi_i(x_i), \tag{10}$$

where the evidence factors are

$$\phi_i(v) = \begin{cases} \mathbf{1}\{v = z_i\}, & i \in \mathcal{R}(z), \\ 1, & i \notin \mathcal{R}(z). \end{cases} \tag{11}$$

Here we take the initial distribution $\pi_0$ to be the stationary distribution of $P'$.

**Forward–backward inference.** To compute posterior marginals efficiently, we apply the standard forward–backward algorithm. Let $\mathcal{E}_{1:i}(z)$ denote the hard-evidence constraints up to position $i$. Define the forward message

$$\alpha_i(v) := p(x_i = v, \mathcal{E}_{1:i}(z)),$$

and the backward message

$$\beta_i(v) := p(\mathcal{E}_{i+1:T}(z) \mid x_i = v).$$

The forward recursion is initialized by

$$\alpha_1(v) = \pi_0(v)\phi_1(v), \qquad v \in \mathcal{V}, \tag{12}$$

and for $i = 2, \dots, T$ is given by

$$\alpha_i(v) = \phi_i(v) \sum_{v' \in \mathcal{V}} \alpha_{i-1}(v') P'(v \mid v'). \tag{13}$$

The backward recursion is initialized by

$$\beta_T(v) = 1, \qquad v \in \mathcal{V}, \tag{14}$$

and for $i = T - 1, \dots, 1$ is given by

$$\beta_i(v) = \sum_{v' \in \mathcal{V}} P'(v' \mid v) \phi_{i+1}(v') \beta_{i+1}(v'). \tag{15}$$

**Oracle denoiser.** The marginal posterior at position $i$ is

$$\gamma_i(v) := p^\star(x_i = v \mid \mathcal{E}(z)) = \frac{\alpha_i(v) \beta_i(v)}{\sum_{v'' \in \mathcal{V}} \alpha_i(v'') \beta_i(v'')}. \tag{16}$$

Here $p^\star(x_i \mid \mathcal{E}(z))$ denotes the *ground-truth oracle posterior marginal* at position $i$, uniquely induced by the known Markov transition kernel $P'$ and the hard evidence $\mathcal{E}(z)$ via exact HMM smoothing. The quantity $\gamma_i$ is its *explicit representation* computed by the forward–backward algorithm.

**Numerical stability.** We implement the forward–backward recursions in the log domain. Let $\ell\alpha_i(v) = \log \alpha_i(v)$ and $\ell\beta_i(v) = \log \beta_i(v)$. Then the smoothing marginal is

$$\log \gamma_i(v) = \ell\alpha_i(v) + \ell\beta_i(v) \\ - \text{logsumexp}_{u \in \mathcal{V}} \left( \ell\alpha_i(u) + \ell\beta_i(u) \right). \tag{17}$$

Derivation and implementation details for HMM forward–backward inference are provided in Appendix A.

# 4. Oracle Instantiations of dLLM Samplers

Having defined the oracle posterior in Section 3.2, we now specify how the same posterior is used inside different reverse samplers. For a current masked state $z_t$, each sampler receives the oracle marginal $\gamma_{t,u}$ in place of its learned denoiser output. Thus, any difference in behavior comes from the sampler update rule rather than from denoiser approximation error.

We instantiate this oracle replacement for SEDD, MDLM, LLaDA, and ReMDM below.

## 4.1. SEDD Sampler: Oracle Scores

SEDD (Lou et al., 2023) uses score ratios rather than direct token probabilities. For a masked position $u$, we obtain the oracle score from the HMM posterior marginal $\gamma_{t,u}$:

$$s^\star(z_t, t)_{u,v} \propto \gamma_{t,u}(v), \qquad v \in \mathcal{V}. \tag{18}$$

Thus, the learned score model in SEDD is replaced by the exact oracle score induced by the ground-truth Markov chain. The full derivation is given in Appendix B.

**Temperature-induced score sharpening in SEDD.** Motivated by observations that low-precision Gumbel-based categorical sampling can induce low-temperature, overconfident behavior (Zheng et al., 2024), we simulate this effect in SEDD by temperature-scaling the oracle score:

$$\log s^{(\beta)}(z_t, t)_{u,v} \propto \beta \log \gamma_{t,u}(v), \qquad \beta \geq 1. \tag{19}$$

Here $\beta = 1$ gives the exact oracle SEDD score, while $\beta > 1$ artificially induces a lower-temperature score distribution without changing token rankings. This provides a controlled way to introduce score-level sampling distortion while keeping the oracle posterior fixed.

### 4.2. MDLM and LLaDA Samplers: Parallel Oracle Decoding

MDLM (Sahoo et al., 2024) and LLaDA (Nie et al., 2026) update multiple masked positions in parallel. We replace their learned denoisers with the factorized oracle marginals

$$p_\theta(x_0 \mid z_t) \;\mapsto\; \prod_{u=1}^{T} \gamma_{t,u}(\cdot). \tag{20}$$

MDLM samples selected masked positions from the oracle marginals, where the selected positions are determined by a random unmasking schedule.

LLaDA uses the same oracle marginals, but selects positions through confidence-based parallel remasking.

### 4.3. ReMDM Sampler: Oracle-Guided Re-masking

ReMDM (Wang et al., 2026) extends masked diffusion by allowing decoded tokens to be re-masked and re-sampled during generation. Following Wang et al. (2026), we evaluate two re-masking variants with the learned denoiser replaced by the oracle posterior.

**ReMDM-conf.** Uses confidence-based re-masking. In the oracle setting, token confidence is computed from the oracle marginal, and lower-confidence tokens are re-masked more frequently.

**ReMDM-loop.** Uses a loop-based schedule in which re-masking is active only within a fixed diffusion-time window. Outside this window, the sampler reduces to standard MDLM updates.

**Nucleus sampling.** Following Wang et al. (2026), we optionally apply nucleus (top-$p$) (Holtzman et al., 2019) sampling when drawing from the oracle marginals. This top-$p$ truncation is a sampler-side decoding choice and is distinct from the top-$k$ truncation used to define the sparse ground-truth transition kernel $P'$.

Further details are provided in Appendix C.4.

**Summary.** We preserve each sampler's original logic and replace only its learned denoiser. ReMDM top-$p$ sampling follows the original sampler, while SEDD temperature scaling is used only as a sensitivity perturbation.

## 5. Metrics

We group our evaluation metrics into four categories. Transition-level metrics are our primary correctness measures because they compare generated token-to-token statistics directly against the oracle transition kernel $P'$. Complete definitions are provided in Appendix D.

**Transition-level correctness.** We measure agreement between the empirical transition distribution $\hat{p}(x_{t+1} \mid x_t)$ and the oracle transition kernel $P'$. We report Sequence NLL (per token), Transition KL, and Transition Entropy. Lower Sequence NLL and Transition KL indicate closer agreement with $P'$, while Transition Entropy diagnoses over-sharpening or excessive randomness.

**Sequence-level diversity.** We report Duplication rate and $n$-gram diversity ratios (2/3-gram) to summarize surface-level diversity. These metrics do not measure distributional correctness, but help characterize qualitative generation behavior.

**Coverage and collapse.** We track coverage diagnostics such as the fraction of generated transitions outside the high-probability support of $P'$ and the effective support size. This support is defined only for evaluation and is distinct from sampler-side nucleus sampling.

**External language-model metrics.** We report GenPPL and MAUVE using a pretrained GPT-2 Large evaluator. Lower GenPPL and higher MAUVE are conventionally interpreted as better, but we treat them as auxiliary diagnostics rather than primary correctness metrics.

## 6. Experimental Setup

Across all experiments, we fix the sequence length $T = 1024$, sample $N = 512$ sequences per setting, and vary the number of diffusion steps $S \in \{8, 16, 32, 64, 128, 256, 512, 1024\}$.

**Datasets and Oracle Transition Kernels.** We primarily evaluate on token-level OpenWebText (OWT) (Gokaslan et al., 2019). For computational tractability, we tokenize OWT with a ByteBPE vocabulary of size $V = 4096$ and construct a bigram oracle transition model from the resulting token sequences. We sparsify this transition kernel by retaining, for each state, enough outgoing transitions to preserve at least $99\%$ cumulative mass; the global cutoff $K$ is set to the 90th percentile of these per-state effective supports, yielding $K = 206$. We then add a small teleport probability $\varepsilon = 10^{-4}$.

For completeness, we also report supplementary character-level Text8 experiments (Sukhbaatar et al., 2015), using a dense bigram prior. Full dataset, tokenizer, and oracle-construction details are provided in Appendix E.

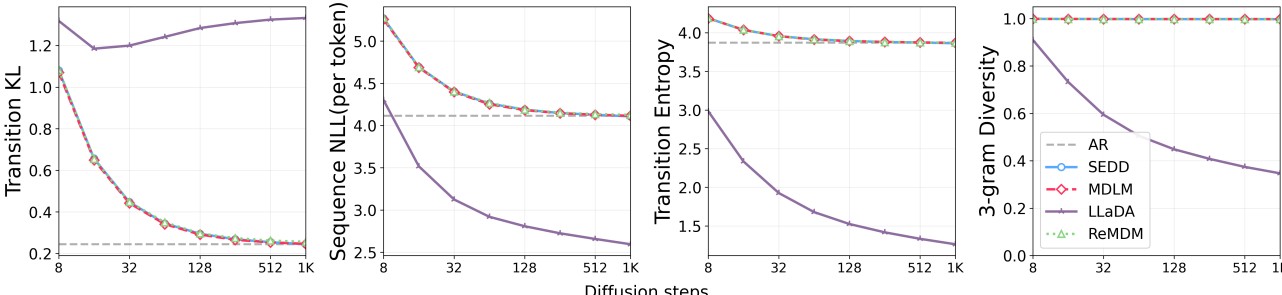

*Figure 2.* **Transition-level metrics on OpenWebText (OWT) under an oracle denoiser.** We report transition KL, NLL, entropy rate, and $n$-gram diversity as functions of the number of sampling steps for several discrete diffusion samplers, using an exact oracle posterior to isolate sampler-induced error. For all samplers, substantial transition-level error persists at small step counts, with convergence to the autoregressive baseline occurring only when the number of steps approaches the sequence length $T$ (except LLaDA). Notably, LLaDA attains very low sequence NLL at few steps, despite severe degradation in transition KL, entropy, and diversity, demonstrating that NLL alone is not a reliable indicator of distributional correctness. All samplers show zero exact duplication.

**Sampler settings.** All samplers use the same oracle posterior while varying the number of diffusion steps. For SEDD, we evaluate $\beta = 1$ and sharpened scores with $\beta > 1$. For ReMDM, we follow the original sampler settings (Wang et al., 2026) and evaluate confidence-based and loop-based re-masking, with and without nucleus sampling ($p = 0.9$). We set $\eta_{\text{cap}} = 0.02$; for the loop variant, $t_{\text{on}} = 0.55$, $t_{\text{off}} = 0.05$, and $\alpha(t_{\text{on}}) = 0.9$.

## 7. Experimental Results and Analysis

We structure the empirical analysis around three main findings. First, sampler-induced error persists even when denoiser approximation error is removed. Section 7.1 shows that few-step dLLM samplers remain far from the oracle transition law, despite using exact oracle marginals.

Second, commonly used external metrics can obscure this error. Sections 7.2 and 7.3 show that GenPPL can be improved by local sharpening alone, while Section 7.4 shows that MAUVE can remain high despite substantial transition-level mismatch.

Third, oracle replacement reveals a sampler–denoiser coupling in LLaDA. Section 7.5 shows that LLaDA is especially sensitive to replacing learned denoiser confidence with the HMM oracle posterior, suggesting that its behavior depends on the interaction between the sampler and the learned denoiser rather than on the sampler rule alone.

### 7.1. Sampler-induced error persists in the few-step regime

Figure 2 shows that, on OpenWebText under an exact oracle denoiser, all evaluated discrete diffusion samplers exhibit substantial transition-level mismatch at small numbers of sampling steps. Across transition KL, NLL, and entropy, agreement with the autoregressive baseline improves only

as the number of sampling steps approaches the sequence length $T$ (except for LLaDA). The same qualitative behavior holds on the character-level Text8 dataset despite the different vocabulary and granularity; see Figure 7 in Appendix F.2.

**Interpretation.** This result reveals a structural limitation of few-step discrete diffusion sampling. These samplers update multiple positions in parallel using marginal token predictions. However, matching local marginals does not generally imply matching the joint sequence distribution, except under strong independence assumptions that are violated in natural language. As a result, transition-level inconsistencies persist in the few-step regime and only diminish when the number of refinement steps becomes comparable to the sequence length. Thus, even with an exact oracle denoiser, few-step dLLM samplers do not faithfully recover the true transition law. See Appendix F.1 for all metric outputs.

### 7.2. Score sharpening improves GenPPL while degrading transition correctness

Figure 3 shows that increasing the sharpening factor $\beta$ ($\beta > 1$) substantially increases transition KL and reduces 3-gram diversity, indicating growing deviation from the true transition distribution. In contrast, sequence NLL and GenPPL decrease monotonically with $\beta$, suggesting apparent improvement under low-temperature sampling. At the same time, MAUVE drops sharply, revealing severe degradation in distributional similarity.

**Interpretation.** Temperature scaling concentrates probability mass on fewer high-density patterns. This locally improves likelihood-based metrics such as NLL and GenPPL, while distorting the true transition law, as reflected by higher KL and reduced diversity. The sharp decline in MAUVE highlights its sensitivity to low-temperature collapse. Overall, these results show that GenPPL can misleadingly im-

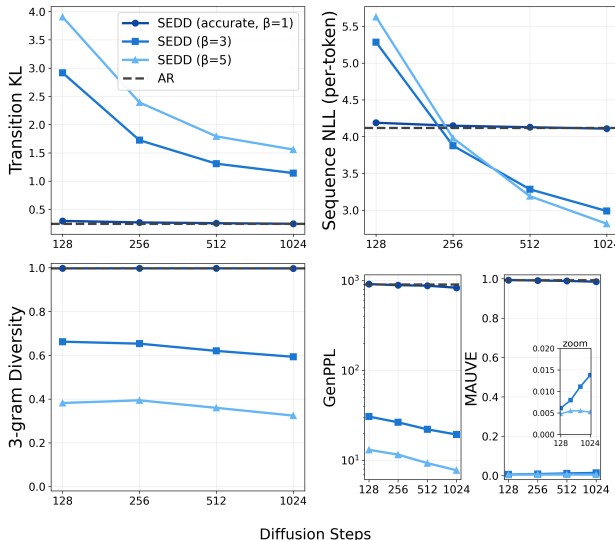

*Figure 3.* **Temperature-induced score sharpening in SEDD under an oracle denoiser (OWT).** As $\beta$ increases, transition KL rises and 3-gram diversity falls, while sequence NLL and GenPPL decrease. MAUVE drops sharply at low temperatures, indicating severe distributional degradation. This shows that likelihood-style metrics can improve under score sharpening even as transition-level correctness and diversity degrade. Duplication remains negligible except under extreme sharpening (Appendix F.2).

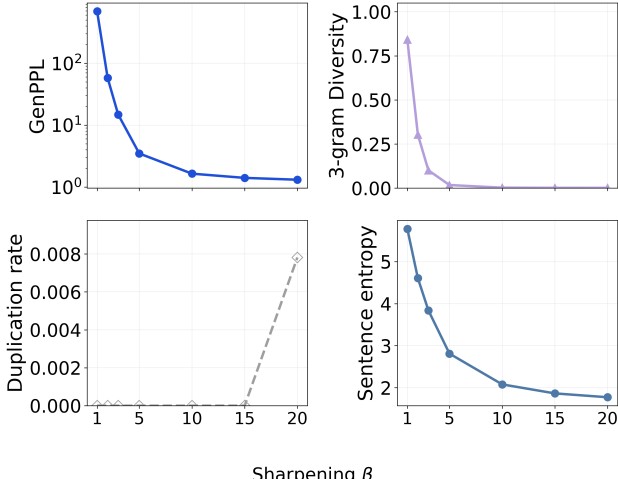

*Figure 4.* **GenPPL under controlled local sharpening in an autoregressive bigram generator (OWT).** We introduce a sharpening factor $\beta$ ($\beta = 1$ corresponds to accurate sampling) and evaluate fixed samples using a pretrained GPT-2 Large model. As $\beta$ increases, GenPPL decreases monotonically, while 3-gram diversity collapses and sentence entropy steadily declines, indicating increasing concentration of probability mass. Exact duplication remains negligible until extreme sharpening.

### 7.3. GenPPL can be gamed by local sharpening alone

To test whether the GenPPL improvement under SEDD temperature scaling is specific to diffusion, we run a minimal autoregressive bigram sanity check. As shown in Figure 4, local sharpening monotonically reduces GenPPL, from 688.4 at accurate sampling ($\beta = 1$) to 1.27 at $\beta = 25$, while simultaneously collapsing $n$-gram diversity and sentence entropy.

This mirrors the SEDD trend despite removing diffusion dynamics and learned denoising entirely. Thus, the GenPPL gain does not require diffusion dynamics or learned denoising: local sharpening alone can reduce GenPPL while destroying diversity and transition faithfulness. Low GenPPL therefore should not be taken as evidence of sampler correctness.

### 7.4. MAUVE is insensitive to transition-level errors

Figure 5 shows that MAUVE remains uniformly high across ReMDM variants and diffusion step counts, even when transition-level metrics vary substantially. In the few-step regime, transition KL, NLL, and entropy indicate clear mismatch from the oracle transition kernel, yet MAUVE changes only mildly. This suggests that MAUVE is rela-

tively insensitive to sampler-induced transition-level error.

The ReMDM variant comparison further illustrates this mismatch. Wang et al. (2026) reports higher MAUVE for ReMDM-loop than ReMDM-conf at large step counts. In our oracle evaluation, MAUVE remains high for both variants with only small differences, whereas transition-level metrics consistently show larger error for ReMDM-loop.

Nucleus sampling shows a related effect. Although Wang et al. (2026) reports that nucleus sampling improves MAUVE, in our oracle setting its main visible effect is on transition-level metrics: transition KL, NLL, and entropy decrease because implausible low-probability tail transitions are truncated. However, MAUVE remains high both with and without nucleus sampling, making this improvement difficult to detect from MAUVE alone.

Taken together, these results show that MAUVE can remain high and nearly unchanged even when sampler variants differ substantially in transition-level correctness. Thus, MAUVE should not be used alone to assess sampler correctness under the oracle transition law.

### 7.5. LLaDA's sampler is coupled to denoiser confidence

LLaDA exhibits a distinct failure mode under oracle replacement (Figure 2). At small step counts, transition NLL appears close to the autoregressive baseline, but transition KL, entropy, and $n$-gram diversity do not converge. Qualitatively, unconditional samples become increasingly template-driven

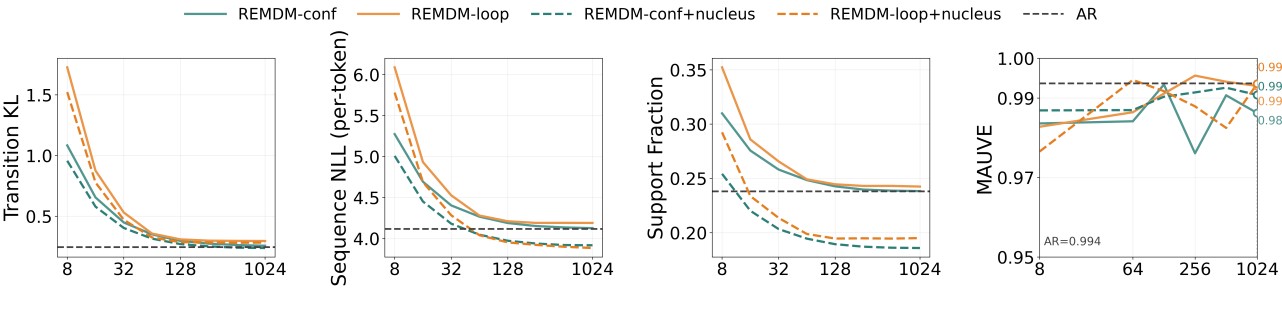

*Figure 5.* **Step-wise evaluation of oracle ReMDM variants on OWT.** Across transition-level metrics, the ReMDM-loop sampler exhibits larger deviations from the AR baseline than ReMDM-conf, indicating higher sampler error, while ReMDM-conf remains consistently closer to the oracle transition kernel. Applying nucleus sampling reduces transition KL, NLL, and entropy, indicating improved alignment with the oracle kernel, but decreases the support fraction due to truncation of low-probability tail transitions. External MAUVE scores remain high and vary only mildly across diffusion steps. A full breakdown of all reported metrics is provided in Appendix Figure 9.

and repetitive as the number of diffusion steps increases, despite showing no exact duplication (Appendix Table 4). We also report ReMDM generations under the same oracle setting as a reference. The contrast suggests that LLaDA is particularly sensitive to replacing the learned denoiser with the HMM oracle posterior (Appendix Table 3).

We attribute this behavior to a strong coupling between LLaDA's sampler and its learned denoiser. LLaDA uses denoiser-produced confidence scores to decide which token predictions to keep and which positions to remask. In the learned model, these confidence scores reflect rich natural-language priors and help shape the remasking dynamics. Under our oracle replacement, the posterior is exact for the first-order Markov target, but it does not contain the same higher-level linguistic structure as the learned denoiser. As a result, the confidence-based selection rule can concentrate probability mass on a small set of high-frequency transition patterns.

This suggests that LLaDA's behavior is not determined by the sampler alone, but by the interaction between the sampler and the confidence structure of the learned denoiser. For confidence-based samplers, denoiser training quality and confidence calibration may therefore play a larger role than in samplers whose updates depend only on oracle marginals or fixed schedules.

## 8. Related Work

**Discrete diffusion language models.** Discrete diffusion language models (dLLMs) enable parallel token updates and provide a non-autoregressive alternative to language modeling (Austin et al., 2021; Li et al., 2022; Lou et al., 2023) and more recently (Sahoo et al., 2024; Chen et al., 2025a; Wang et al., 2026). While these models allow efficient parallel decoding, their sampling procedures introduce additional

sources of approximation beyond model training. Recent work has begun to investigate sampling error in dLLMs (Zheng et al., 2024; Park et al., 2025; Liang et al., 2025; Peng et al., 2025; Zhao et al., 2025; Kang et al., 2025; Chen et al., 2025a), typically evaluating degradation through final generated samples. In such settings, sampler error remains entangled with model approximation error.

**Parallel decoding and independence assumptions.** Several works attribute quality degradation to independence assumptions in parallel updates. Chen et al. (2025a) analyze parallel unmasking under oracle conditionals and derive error bounds based on idealized assumptions. Kang et al. (2025) empirically show that ignoring token dependence leads to quality drops. Bansal & Sanghavi (2025) identify parallel unmasking as product-marginal sampling and propose an auxiliary sampler to approximate joint sampling within a denoising step.

**Evaluation metrics for diffusion language models.** Evaluation of diffusion language models often mixes external generation metrics with downstream task benchmarks. For unconditional generation, prior work has questioned the reliability of metrics such as GenPPL: Zheng et al. (2024) show that conclusions can depend strongly on metric choice, while ReMDM reports MAUVE alongside GenPPL and observes that GenPPL improvements do not consistently translate to downstream quality (Wang et al., 2026). For reasoning-oriented dLLMs, evaluation increasingly relies on zero-shot task performance, including math and coding benchmarks such as GSM8K, MATH500, HumanEval, and MBPP (Nie et al., 2026; Chen et al., 2025b; Feng et al., 2026).

**Optimal denoisers and inference hardness.** Recent theoretical work analyzes sampling in diffusion, flow-based, and autoregressive models from a statistical physics perspective (Ghio et al., 2024), characterizing optimal denoisers and the

hardness of inference in complex graphical models. These analyses show that computing the optimal posterior or score via message passing can become intractable in loopy or high-complexity regimes. Complementary work studies settings in which exact score models can be constructed under specific structural assumptions (Albrychiewicz et al., 2026). While these studies examine the existence and tractability of optimal denoisers, they do not investigate the empirical behavior of sampling algorithms when the denoiser is exact.

**Confidence-based decoding in dLLMs.** Recent work also questions whether confidence-based decoding is always beneficial. Ni et al. (2026) show that arbitrary-order decoding can bypass high-uncertainty tokens and prematurely collapse the reasoning solution space. Fang et al. (2026) further identify a quality–exploration dilemma: low-confidence remasking improves single-sample quality but constrains sequence entropy and limits Pass@$k$ gains. These observations suggest that confidence-based decoding can improve practical sample quality by avoiding uncertain positions, but it also makes the sampler more dependent on the denoiser's confidence structure.

## 9. Conclusion

We introduced an oracle-based framework that isolates sampler-induced error in discrete diffusion language models by replacing learned denoisers with exact ground-truth posteriors. In this setting, few-step samplers remain substantially biased: transition-level mismatch persists even under oracle denoising and only diminishes as the number of sampling steps approaches the sequence length.

We further show that common external metrics can obscure this error. GenPPL can improve under local sharpening without faithful sampling, while MAUVE can remain high despite transition-level deviations. LLaDA further shows that samplers using denoiser confidence for token selection can be tightly coupled to the learned denoiser's confidence structure. Overall, our results motivate transition-aware evaluation that separates sampler behavior from denoiser quality and surface-level metrics.

More broadly, our findings align with concurrent efforts to evaluate diffusion language models at the level of the test-time sampler. Concurrent work (Turok et al., 2026) proposes deterministic unmasking policies with exact likelihoods and AR-like perplexity scores. Our results address a complementary question for parallel and stochastic samplers: whether their transition-level behavior is faithful to the target reverse process. Together, these perspectives build toward the eventual goal of dLLM samplers that are both fast and verifiably faithful to the target distribution.

## Impact Statement

This work contributes a methodological framework for evaluating discrete diffusion language model samplers under controlled oracle settings. It does not introduce a new deployed model, collect sensitive data, or involve human-subject experiments. We do not anticipate direct societal risks beyond those generally associated with advances in generative modeling.

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

# A. Derivation of the Oracle Posterior via Forward–Backward

This appendix provides a detailed derivation of the oracle posterior $p(x_t \mid \mathcal{E}(z))$ for a discrete Markov-chain ground-truth model under partial hard-evidence observations. We review the standard forward–backward algorithm for hidden Markov models, derive the forward and backward recursions explicitly, and then specialize the result to deterministic masking observations and to the rank-one teleport transition used in our experiments. We conclude with a numerically stable log-domain implementation.

## A.1. Hidden Markov Model Setup

Let $x_{1:T} = (x_1, \ldots, x_T) \in \mathcal{V}^T$ denote a discrete-time Markov chain with initial distribution $\pi_0$ and transition kernel $P'$,

$$p_0(x_{1:T}) = \pi_0(x_1) \prod_{t=2}^{T} P'(x_t \mid x_{t-1}). \tag{21}$$

We consider partial observations $z_{1:T} \in (\mathcal{V} \cup \{\texttt{MASK}\})^T$, where $z_t = \texttt{MASK}$ indicates that the clean token $x_t$ is unobserved. This induces an HMM view in which $x_t$ are latent states and $z_t$ provides hard evidence at revealed positions.

Define the set of revealed positions

$$\mathcal{R}(z) := \{t \in \{1, \ldots, T\} : z_t \neq \texttt{MASK}\},$$

and the corresponding hard-evidence event

$$\mathcal{E}(z) := \{x_{1:T} \in \mathcal{V}^T : x_t = z_t, \ \forall t \in \mathcal{R}(z)\}.$$

The inference task of interest is the smoothing posterior

$$p(x_t = v \mid \mathcal{E}(z)), \qquad t = 1, \ldots, T. \tag{22}$$

## A.2. Forward Recursion (Derivation)

Let $\mathcal{E}_{1:t}(z)$ denote the hard-evidence constraints up to position $t$. Define the forward message

$$\alpha_t(v) := p(x_t = v, \mathcal{E}_{1:t}(z)), \qquad v \in \mathcal{V}. \tag{23}$$

By the law of total probability,

$$\alpha_t(v) = \sum_{u \in \mathcal{V}} p(x_t = v, \ x_{t-1} = u, \ \mathcal{E}_{1:t}(z)). \tag{24}$$

Applying the chain rule,

$$p(x_t = v, \ x_{t-1} = u, \ \mathcal{E}_{1:t}(z)) = p(\mathcal{E}_t(z) \mid x_t = v) \, p(x_t = v \mid x_{t-1} = u) \, p(x_{t-1} = u, \mathcal{E}_{1:t-1}(z)). \tag{25}$$

Here $p(\mathcal{E}_t(z) \mid x_t = v)$ is the local hard-evidence factor $\phi_t(v)$, and $p(x_{t-1} = u, \mathcal{E}_{1:t-1}(z)) = \alpha_{t-1}(u)$. Therefore,

$$\alpha_t(v) = \phi_t(v) \sum_{u \in \mathcal{V}} \alpha_{t-1}(u) \, P'(v \mid u), \qquad t = 2, \ldots, T. \tag{26}$$

The initialization is

$$\alpha_1(v) = \pi_0(v) \phi_1(v). \tag{27}$$

## A.3. Backward Recursion (Derivation)

Let $\mathcal{E}_{t+1:T}(z)$ denote the hard-evidence constraints after position $t$. Define the backward message

$$\beta_t(v) := p(\mathcal{E}_{t+1:T}(z) \mid x_t = v), \qquad v \in \mathcal{V}. \tag{28}$$

By marginalizing over $x_{t+1}$,

$$\beta_t(v) = \sum_{u \in \mathcal{V}} p(\mathcal{E}_{t+1:T}(z), x_{t+1} = u \mid x_t = v). \tag{29}$$

Applying the chain rule and the Markov property,

$$p(\mathcal{E}_{t+1:T}(z), x_{t+1} = u \mid x_t = v) = p(\mathcal{E}_{t+1:T}(z) \mid x_{t+1} = u)\, p(x_{t+1} = u \mid x_t = v) \tag{30}$$

$$= \phi_{t+1}(u)\, p(\mathcal{E}_{t+2:T}(z) \mid x_{t+1} = u)\, P'(u \mid v). \tag{31}$$

Recognizing $p(\mathcal{E}_{t+2:T}(z) \mid x_{t+1} = u) = \beta_{t+1}(u)$, we obtain

$$\beta_t(v) = \sum_{u \in \mathcal{V}} P'(u \mid v)\, \phi_{t+1}(u)\, \beta_{t+1}(u), \qquad t = T-1, \ldots, 1. \tag{32}$$

The terminal condition is

$$\beta_T(v) = 1. \tag{33}$$

## A.4. Smoothing Marginals

Combining forward and backward messages yields the exact marginal posterior

$$p(x_t = v \mid \mathcal{E}(z)) = \frac{\alpha_t(v)\, \beta_t(v)}{\sum_{u \in \mathcal{V}} \alpha_t(u)\, \beta_t(u)}. \tag{34}$$

## A.5. Hard-Evidence (Masking) Observations

In our setting, each observation is either revealed or masked. A revealed position satisfies $z_t \in \mathcal{V}$, while a masked position satisfies $z_t = \texttt{MASK}$. Hard evidence corresponds to the event

$$\mathcal{E}(z) := \{x_{1:T} \in \mathcal{V}^T : x_t = z_t,\ \forall t \in \mathcal{R}(z)\}.$$

Equivalently, we use evidence factors

$$\phi_t(v) = \begin{cases} \mathbf{1}\{v = z_t\}, & t \in \mathcal{R}(z), \\ 1, & t \notin \mathcal{R}(z). \end{cases} \tag{35}$$

Thus, masked positions provide no discriminative likelihood over the latent state.

With this notation, the posterior over full latent sequences can be written as

$$p(x_{1:T} \mid \mathcal{E}(z)) \propto \pi_0(x_1) \prod_{t=2}^{T} P'(x_t \mid x_{t-1}) \prod_{t=1}^{T} \phi_t(x_t). \tag{36}$$

The forward and backward recursions are exactly Eqs. (26)–(32).

## A.6. Rank-One Teleport Transition and Exact Sparse Message Passing

We use a sparse transition kernel with teleport:

$$P'(v \mid u) = (1 - \varepsilon)\, P_{\text{top-}k}(v \mid u) + \varepsilon\, \nu(v), \qquad \varepsilon \in (0, 1). \tag{37}$$

The teleport term $\nu(v)$ does not depend on the previous state $u$, forming a rank-one mixture. This enables exact message passing without constructing a dense $|\mathcal{V}| \times |\mathcal{V}|$ transition matrix.

**Forward update under $P'$.** Using the hard-evidence factors $\phi_t$, the forward update is

$$\alpha_t(v) = \phi_t(v) \sum_{u \in \mathcal{V}} \alpha_{t-1}(u) P'(v \mid u). \tag{38}$$

Substituting Eq. (37),

$$\sum_{u\in\mathcal{V}}\alpha_{t-1}(u)P'(v\mid u) = (1-\varepsilon)\sum_{u\in\mathcal{V}}\alpha_{t-1}(u)P_{\text{top-}k}(v\mid u) + \varepsilon\,\nu(v)\sum_{u\in\mathcal{V}}\alpha_{t-1}(u). \tag{39}$$

If we keep $\alpha_{t-1}$ normalized, or equivalently track its scale separately, then $\sum_{u\in\mathcal{V}}\alpha_{t-1}(u) = 1$ up to the tracked scale. The prediction step can therefore be written as

$$\widetilde{\alpha}_t(v) = (1-\varepsilon)\sum_{u\in\mathcal{V}}\alpha_{t-1}(u)P_{\text{top-}k}(v\mid u) + \varepsilon\,\nu(v), \qquad \alpha_t(v) = \phi_t(v)\widetilde{\alpha}_t(v), \tag{40}$$

where the first term is computed by sparse accumulation over the top-$k$ edges.

**Backward update under $P'$.** Similarly, the backward update is

$$\beta_t(u) = \sum_{v\in\mathcal{V}} P'(v\mid u)\,\phi_{t+1}(v)\,\beta_{t+1}(v). \tag{41}$$

Substituting Eq. (37) yields

$$\beta_t(u) = (1-\varepsilon)\sum_{v\in\mathcal{V}} P_{\text{top-}k}(v\mid u)\,\phi_{t+1}(v)\,\beta_{t+1}(v) + \varepsilon\sum_{v\in\mathcal{V}}\nu(v)\,\phi_{t+1}(v)\,\beta_{t+1}(v). \tag{42}$$

Define the global scalar, independent of $u$,

$$c_{t+1} := \sum_{v\in\mathcal{V}}\nu(v)\,\phi_{t+1}(v)\,\beta_{t+1}(v). \tag{43}$$

Then

$$\beta_t(u) = (1-\varepsilon)\sum_{v\in\mathcal{V}} P_{\text{top-}k}(v\mid u)\,\phi_{t+1}(v)\,\beta_{t+1}(v) + \varepsilon\,c_{t+1}. \tag{44}$$

Thus, both forward and backward passes remain exact while using only sparse top-$k$ edges plus a rank-one teleport correction.

**Complexity.** If each state stores $k$ outgoing neighbors in $P_{\text{top-}k}$, the sparse accumulations in Eqs. (40) and (44) cost $O(|\mathcal{V}|k)$ per time step, leading to total complexity $O(T|\mathcal{V}|k)$.

### A.7. Log-Domain Implementation (Numerical Stability)

Forward–backward inference involves products of many probabilities and can underflow in finite precision. We therefore implement the recursions in the log domain. Let

$$\ell\alpha_t(v) = \log\alpha_t(v), \quad \ell\beta_t(v) = \log\beta_t(v), \quad \ell\phi_t(v) = \log\phi_t(v),$$

and define

$$\text{LSE}_{u\in\mathcal{V}}\,f(u) := \log\sum_{u\in\mathcal{V}}\exp(f(u)).$$

**Hard evidence in log domain.** The hard-evidence factors become

$$\ell\phi_t(v) = \begin{cases} 0, & t\notin\mathcal{R}(z), \\ 0, & t\in\mathcal{R}(z)\text{ and }v = z_t, \\ -\infty, & t\in\mathcal{R}(z)\text{ and }v\neq z_t. \end{cases} \tag{45}$$

**Log-domain forward recursion with rank-one teleport.** Assuming the forward messages are normalized at each step, define

$$s_t(v) := \sum_{u\in\mathcal{V}}\alpha_{t-1}(u)P_{\text{top-}k}(v\mid u), \qquad \widetilde{\alpha}_t(v) = (1-\varepsilon)s_t(v) + \varepsilon\nu(v).$$

Writing $\ell s_t(v) = \log s_t(v)$ and $\ell \nu(v) = \log \nu(v)$, we compute

$$\log \widetilde{\alpha}_t(v) = \mathrm{LSE}\left(\log(1-\varepsilon) + \ell s_t(v),\ \log \varepsilon + \ell \nu(v)\right). \tag{46}$$

Then we apply evidence and normalize:

$$\ell \alpha_t(v) = \ell \phi_t(v) + \log \widetilde{\alpha}_t(v) - \log Z_t, \qquad \log Z_t = \mathrm{LSE}_{u \in \mathcal{V}}\left(\ell \phi_t(u) + \log \widetilde{\alpha}_t(u)\right). \tag{47}$$

**Log-domain backward recursion with rank-one teleport.** Define

$$a_t(u) := \sum_{v \in \mathcal{V}} P_{\text{top-}k}(v \mid u)\, \phi_{t+1}(v)\, \beta_{t+1}(v), \qquad c_{t+1} := \sum_{v \in \mathcal{V}} \nu(v)\, \phi_{t+1}(v)\, \beta_{t+1}(v).$$

Then

$$\beta_t(u) = (1-\varepsilon)a_t(u) + \varepsilon c_{t+1}.$$

We compute

$$\ell c_{t+1} = \mathrm{LSE}_{v \in \mathcal{V}}\left(\ell \nu(v) + \ell \phi_{t+1}(v) + \ell \beta_{t+1}(v)\right), \tag{48}$$

and, with $\ell a_t(u) = \log a_t(u)$, combine the two mixture terms as

$$\ell \beta_t(u) = \mathrm{LSE}\left(\log(1-\varepsilon) + \ell a_t(u),\ \log \varepsilon + \ell c_{t+1}\right) - \log \widetilde{C}_t. \tag{49}$$

Here $\log \widetilde{C}_t$ is an arbitrary per-time additive constant, optionally used for normalization, since scaling $\beta_t$ does not change the final posterior marginal.

**Smoothing marginal in log domain.** Finally,

$$\log \gamma_t(v) = \ell \alpha_t(v) + \ell \beta_t(v) - \mathrm{LSE}_{u \in \mathcal{V}}\left(\ell \alpha_t(u) + \ell \beta_t(u)\right), \qquad \gamma_t(v) = \exp(\log \gamma_t(v)). \tag{50}$$

## A.8. Log-Domain Implementation (Numerical Stability)

Forward–backward inference involves products of many probabilities and can underflow in finite precision. We therefore implement the recursions in the log domain. Let

$$\ell \alpha_t(v) = \log \alpha_t(v), \quad \ell \beta_t(v) = \log \beta_t(v), \quad \ell \phi_t(v) = \log \phi_t(v),$$

and define

$$\mathrm{LSE}_{u \in \mathcal{V}}\, f(u) = \log \sum_{u \in \mathcal{V}} \exp(f(u)).$$

**Hard evidence in log domain.** The hard-evidence factors become

$$\ell \phi_t(v) = \begin{cases} 0, & t \notin \mathcal{R}(z), \\ 0, & t \in \mathcal{R}(z) \text{ and } v = z_t, \\ -\infty, & t \in \mathcal{R}(z) \text{ and } v \neq z_t. \end{cases} \tag{51}$$

**Log-domain forward recursion with rank-one teleport.** Recall that

$$P'(v \mid u) = (1-\varepsilon)P_{\text{top-}k}(v \mid u) + \varepsilon \nu(v).$$

Assuming the forward messages are normalized at each step, define

$$s_t(v) := \sum_{u \in \mathcal{V}} \alpha_{t-1}(u) P_{\text{top-}k}(v \mid u), \qquad \widetilde{\alpha}_t(v) = (1-\varepsilon)s_t(v) + \varepsilon \nu(v).$$

Writing $\ell s_t(v) = \log s_t(v)$ and $\ell \nu(v) = \log \nu(v)$, we compute

$$\log \widetilde{\alpha}_t(v) = \mathrm{LSE}\left(\log(1-\varepsilon) + \ell s_t(v),\ \log \varepsilon + \ell \nu(v)\right). \tag{52}$$

Then we apply evidence and normalize:

$$\ell\alpha_t(v) = \ell\phi_t(v) + \log\widetilde{\alpha}_t(v) - \log Z_t, \qquad \log Z_t = \mathrm{LSE}_{u\in\mathcal{V}}\left(\ell\phi_t(u) + \log\widetilde{\alpha}_t(u)\right). \tag{53}$$

**Log-domain backward recursion with rank-one teleport.** Define

$$a_t(u) := \sum_{v\in\mathcal{V}} P_{\text{top-}k}(v\mid u)\phi_{t+1}(v)\beta_{t+1}(v), \qquad c_{t+1} := \sum_{v\in\mathcal{V}} \nu(v)\phi_{t+1}(v)\beta_{t+1}(v).$$

Then

$$\beta_t(u) = (1-\varepsilon)a_t(u) + \varepsilon c_{t+1}.$$

We compute

$$\ell c_{t+1} = \mathrm{LSE}_{v\in\mathcal{V}}\left(\ell\nu(v) + \ell\phi_{t+1}(v) + \ell\beta_{t+1}(v)\right), \tag{54}$$

and, with $\ell a_t(u) = \log a_t(u)$, combine the two mixture terms as

$$\ell\beta_t(u) = \mathrm{LSE}\left(\log(1-\varepsilon) + \ell a_t(u),\ \log\varepsilon + \ell c_{t+1}\right) - \log\widetilde{C}_t. \tag{55}$$

Here $\log\widetilde{C}_t$ is an arbitrary per-time additive constant, optionally used for normalization, since scaling $\beta_t$ does not change the final posterior marginal.

**Smoothing marginal in log domain.** Finally,

$$\log\gamma_t(v) = \ell\alpha_t(v) + \ell\beta_t(v) - \mathrm{LSE}_{u\in\mathcal{V}}\left(\ell\alpha_t(u) + \ell\beta_t(u)\right), \qquad \gamma_t(v) = \exp(\log\gamma_t(v)). \tag{56}$$

# B. Derivation of the Oracle Concrete Score under a Markov Prior (SEDD)

## B.1. Forward Noising Model

Let $x_0 = (x_{0,1},\ldots,x_{0,T}) \in \mathcal{V}^T$ be a clean sequence drawn from a Markov chain prior

$$p_0(x_0) = \pi_0(x_{0,1}) \prod_{t=2}^{T} P(x_{0,t}\mid x_{0,t-1}). \tag{57}$$

Following SEDD, we consider the absorbing (MASK) forward process. At time $t$, each position is independently corrupted according to

$$p_{t|0}(z_u = x_{0,u}\mid x_{0,u}) = e^{-\sigma(t)}, \qquad p_{t|0}(z_u = \texttt{MASK}\mid x_{0,u}) = 1 - e^{-\sigma(t)}. \tag{58}$$

This induces a noisy marginal $p_t(z) = \sum_{x_0} p_0(x_0)p_{t|0}(z\mid x_0)$.

## B.2. Concrete Score in SEDD

For a noisy sequence $z$ with $z_u = \texttt{MASK}$, define

$$x = (z_{-u}, \texttt{MASK}), \qquad y = (z_{-u}, i), \quad i\in\mathcal{V}.$$

The concrete score required by SEDD is

$$s_t(z,u,i) \triangleq \frac{p_t(y)}{p_t(x)} = \frac{p_t(z_u = i, z_{-u})}{p_t(z_u = \texttt{MASK}, z_{-u})}. \tag{59}$$

## B.3. Score–Posterior Identity

Using the absorbing channel,

$$p_t(z_u = \texttt{MASK}, z_{-u}) = \left(1 - e^{-\sigma(t)}\right)p_t(z_{-u}), \tag{60}$$

$$p_t(z_u = i, z_{-u}) = e^{-\sigma(t)}\, p(x_{0,u} = i, z_{-u}). \tag{61}$$

Taking the ratio yields

$$s_t(z,u,i) = \frac{e^{-\sigma(t)}}{1 - e^{-\sigma(t)}}\, p(x_{0,u} = i\mid z). \tag{62}$$

### B.4. Oracle SEDD

Thus, under a known Markov prior, the concrete score required by SEDD is given exactly by a scaled posterior marginal. Replacing the neural approximation with this exact score yields an oracle SEDD that preserves the original reverse-time dynamics while eliminating denoiser error.

## C. Oracle Instantiations of Discrete Diffusion Samplers

### C.1. Samplers Under Comparison

We compare a set of representative discrete sequence samplers, ranging from a ground-truth autoregressive baseline to several masked diffusion–style models. Our primary goal is to isolate and analyze *sampling error* by removing denoiser error wherever possible.

**Baseline: autoregressive sampling.** As a reference point, we include an autoregressive (AR) sampler that draws sequences directly from the ground-truth Markov chain. This sampler exactly follows the true transition kernel and therefore incurs no sampling error by construction. It serves as an upper bound on achievable performance under the given ground truth.

**Masked diffusion samplers.** We evaluate four families of masked diffusion samplers: **SEDD**, **MDLM**, **ReMDM**, and **LLaDA**. Although these methods differ substantially in their reverse-time dynamics, they all share a common high-level structure: at each iteration, a denoiser produces local conditional information about clean tokens given a partially specified sequence, and a sampler uses this information to update the latent state via masking, remasking, or parallel token prediction.

**Oracle substitution and controlled evaluation.** In the original models, the denoiser is implemented as a trained neural network. In our controlled setting, we replace the neural denoiser with an *oracle posterior* computed exactly under the ground-truth Markov prior. Specifically, given the current latent state, we compute exact conditional marginals via HMM forward–backward inference with hard evidence at unmasked positions. This substitution removes denoiser error entirely while preserving the original sampler logic, including update order, remasking strategy, and stochasticity.

**Accurate and inaccurate oracle variants.** For SEDD, we further introduce an *inaccurate oracle* variant that applies a temperature scaling to the concrete score at sampling time. This controlled perturbation leaves the oracle posterior intact but deliberately distorts how the sampler consumes the score, providing a diagnostic setting to study the sensitivity of the sampling mechanism itself. Comparing accurate and inaccurate oracle SEDD allows us to directly assess the impact of sampling miscalibration in the absence of denoiser error.

**Organization.** Below, we describe each sampler in turn. For each method, we first summarize the form of the neural denoiser output in the original algorithm, and then explain how it can be replaced by the exact oracle posterior while keeping the sampling dynamics unchanged.

### C.2. SEDD: Concrete Score and Oracle Replacement

**Score-based parameterization in SEDD.** Score-Entropy Discrete Diffusion Models (SEDD) (Lou et al., 2023) define a continuous-time Markov jump process over discrete sequences with an absorbing `[MASK]` state. Let $p_t$ denote the marginal distribution of the noisy sequence at time $t$. The reverse-time dynamics are parameterized by *concrete score ratios* between discrete states, which determine the token-level jump rates.

Under the absorbing noise model, the only nontrivial reverse transitions occur from `[MASK]` to vocabulary tokens. For a position $u$ with $z_u = $ `[MASK]`, the reverse jump rate to token $i \in \mathcal{V}$ depends on the ratio $p_t(x_{0,u} = i \mid z)$.

**Neural approximation of the concrete score.** In practice, SEDD employs a neural network that outputs a categorical distribution

$$\hat{p}_\theta(x_{0,u} = i \mid z), \tag{63}$$

which is not used directly by the sampler. Instead, it is converted into a concrete score of the form

$$s_t^\theta(z, u, i) = \frac{e^{-\sigma(t)}}{1 - e^{-\sigma(t)}} \, \hat{p}_\theta(x_{0,u} = i \mid z), \tag{64}$$

where $\sigma(t)$ denotes the continuous-time noise schedule. The sampler uses these scores to define reverse jump intensities from `[MASK]` to token states.

**Sampler-side temperature via log-score scaling.** To probe the sensitivity of SEDD to score miscalibration, we introduce a sampler-side temperature that rescales the *log concrete scores*. Given any score field $s_t(z_t, u, i)$ (neural or oracle), we form a tempered score

$$\log s_t^{(\beta)}(z_t, u, i) \;=\; \beta \, \log s_t(z_t, u, i), \qquad \beta > 0, \tag{65}$$

equivalently,

$$s_t^{(\beta)}(z_t, u, i) \;=\; \left( s_t(z_t, u, i) \right)^{\beta}. \tag{66}$$

When $\beta > 1$, the score becomes sharper (more mass on high-score tokens), whereas $\beta < 1$ flattens the score. In our "inaccurate oracle SEDD" ablation, we apply (65) to the oracle score in (69) while keeping the $\tau$-leaping sampler unchanged.

**Exact oracle posterior under a known Markov prior.** We assume access to a ground-truth generative model $p_0(x_0)$ given by a known Markov chain, together with the same absorbing forward noising process as in SEDD. Conditioned on a noisy sequence $z_t$, we define the *oracle denoiser* as the exact posterior over clean sequences,

$$p^{\star}(x_0 \mid z_t) \;\triangleq\; p(x_0 \mid z_t), \tag{67}$$

which can be computed exactly via HMM forward–backward smoothing with hard evidence at unmasked positions.

**Oracle concrete score induced by** $p^{\star}(x_0 \mid z_t)$**.** SEDD parameterizes reverse-time dynamics through token-level concrete scores rather than directly through the posterior. For a position $u$ with $z_u = \texttt{[MASK]}$, the reverse jump rate to token $i \in \mathcal{V}$ depends on a local concrete score ratio of the form $p_t((z_{-u}, i))/p_t((z_{-u}, \texttt{[MASK]}))$, which under the absorbing model reduces to a known time-dependent factor times a single-site posterior marginal. Accordingly, for a position $u$ and token $i \in \mathcal{V}$, we define

$$\gamma_u(i) \;\triangleq\; p^{\star}(x_{0,u} = i \mid z_t), \tag{68}$$

as the marginal of the oracle posterior at position $u$.

Crucially, under the absorbing noise process, the exact concrete score induced by the oracle posterior admits the closed form

$$s_t(z_t, u, i) = \frac{e^{-\sigma(t)}}{1 - e^{-\sigma(t)}} \, \gamma_u(i), \tag{69}$$

which is exactly the quantity required by the SEDD sampler.

**Tweedie $\tau$-leaping sampler.** SEDD performs sampling using a Tweedie-style $\tau$-leaping approximation of the reverse-time continuous-time Markov jump process. Over a finite step size $\Delta t$, token updates are applied independently and simultaneously using the concrete scores $s_t(z_t, u, i)$. When the score is exact, this $\tau$-leaping scheme is optimal among all independent token update rules (Lou et al., 2023).

**Oracle SEDD.** We construct an *oracle SEDD* sampler by substituting the exact concrete scores induced by $p^{\star}(x_0 \mid z_t)$ in (69) for the neural approximation used in standard SEDD, while keeping the continuous-time dynamics and $\tau$-leaping discretization unchanged.

### C.3. MDLM: Masked Diffusion without Remasking

Masked Diffusion Language Models (MDLM) (Sahoo et al., 2024) constitute the simplest class of discrete masked diffusion samplers and serve as the conceptual foundation for several later variants, including ReMDM and LLaDA.

**Forward process and latent state.** MDLM defines a forward noising process in which tokens are independently replaced by a special mask symbol $m = \texttt{[MASK]}$ according to a predefined noise schedule $\{\alpha_t\}_{t=0}^{T}$. At reverse time $t$, the latent state

$$z_t \in (\mathcal{V} \cup \{m\})^T$$

consists of a partially masked sequence, with masked positions treated as latent variables.

**Reverse-time posterior and sampling.** Given a clean sequence $x$, the MDLM reverse-time posterior corresponds to the special case of ReMDM with no remasking, i.e., $\sigma_t = 0$. For masked positions, the reverse kernel reduces to drawing tokens from the posterior marginal,

$$p(z_{t-1}^{(i)} \mid z_t, x) = \text{Cat}\left( z_{t-1}^{(i)}; x^{(i)} \right), \qquad \text{if } z_t^{(i)} = m, \tag{70}$$

while unmasked positions remain fixed. In practice, the clean sequence $x$ is unknown and replaced by the output of a denoiser.

**Neural denoiser and oracle substitution.** In standard MDLM, a neural network predicts token-wise conditionals

$$p_\theta(x_i \mid z_t),$$

which are used to sample masked positions in parallel at each reverse step.

In our controlled setting, we replace the neural denoiser with the exact oracle posterior

$$p^\star(x \mid z_t),$$

computed via HMM forward–backward inference under the ground-truth Markov prior, with unmasked positions treated as hard evidence. The oracle-MDLM reverse update therefore samples masked tokens independently according to

$$z_{t-1}^{(i)} \sim \mathrm{Cat}(p^\star(x_i \mid z_t)), \qquad \text{for all } i \text{ with } z_t^{(i)} = m. \tag{71}$$

### C.4. ReMDM: Oracle-Guided Re-masking

**ReMasking Diffusion Models (ReMDM).** ReMasking Diffusion Models (ReMDM) (Wang et al., 2026) extend masked diffusion models by allowing previously decoded tokens to be re-masked and re-sampled during reverse-time generation. Let $z_t \in (\mathcal{V} \cup \{m\})^T$ denote the latent sequence at time $t$, with $m = \texttt{[MASK]}$. Conditioned on a clean sequence $x$, ReMDM defines a reverse-time posterior that generalizes MDLM by introducing a re-masking probability $\sigma_t$.

**Reverse-time posterior.** Let $s < t$ denote the previous timestep. Conditioned on $x$, the reverse kernel (Eq. (6) of the original paper) is

$$p(z_s \mid z_t, x) = \begin{cases} \mathrm{Cat}(z_s; \, (1 - \sigma_t)x + \sigma_t m), & z_t \neq m, \\ \mathrm{Cat}\left( z_s; \, \dfrac{\alpha_s - (1 - \sigma_t)\alpha_t}{1 - \alpha_t} x \right. \\ \left. \qquad + \dfrac{1 - \alpha_s - \sigma_t \alpha_t}{1 - \alpha_t} m \right), & z_t = m, \end{cases} \tag{72}$$

where $\alpha_t$ is the forward noising schedule. Setting $\sigma_t = 0$ recovers standard MDLM.

During sampling, $x$ is replaced by the output of a denoiser, yielding an approximate reverse kernel.

**Oracle substitution.** In standard ReMDM, a neural denoiser $x_\theta(z_t)$ approximates $p(x \mid z_t)$. In our controlled setting, we replace the neural denoiser with the exact oracle posterior

$$p^\star(x \mid z_t),$$

computed via HMM forward–backward inference under the ground-truth Markov model. All other components—posterior form (72), noise schedule $\alpha_t$, and re-masking schedule—remain unchanged. Any deviation from the ground-truth distribution therefore reflects sampler-induced error alone.

**ReMDM-loop.** The loop-based schedule activates re-masking only within a diffusion-time window

$$t_{\mathrm{off}} < t \leq t_{\mathrm{on}}.$$

Outside this window, the sampler reduces to standard MDLM updates. Within the active window, the re-masking rate follows a capped schedule

$$\sigma_t = \min\{\eta_{\mathrm{cap}}, \sigma_t^{\max}\}, \tag{73}$$

where $\eta_{\mathrm{cap}}$ limits the re-masking probability and $\sigma_t^{\max}$ ensures validity of the reverse posterior.

**ReMDM-conf.** The confidence-based schedule modulates re-masking probabilities at the token level based on denoiser confidence. Let $\ell$ index sequence positions. For unmasked tokens, the oracle confidence is defined as

$$\psi_t^{(\ell)} := p^\star(x_\ell = z_t^{(\ell)} \mid z_t), \tag{74}$$

with $\psi_t^{(\ell)} = \infty$ if $z_t^{(\ell)} = m$. The per-token re-masking probability becomes

$$\sigma_t^{(\ell)} = \eta_{\text{conf}}^{(\ell)} \cdot \sigma_t, \qquad \eta_{\text{conf}}^{(\ell)} = \frac{\exp(-\psi_t^{(\ell)})}{\sum_j \exp(-\psi_t^{(j)})}. \tag{75}$$

Tokens with lower oracle confidence are therefore more likely to be re-masked.

**Evaluation protocol.** On large-scale benchmarks (e.g., OWT), we report results for **ReMDM-conf** as the primary ReMDM variant. On smaller benchmarks, we additionally include **ReMDM-loop** to isolate the effect of windowed re-masking.

### C.5. LLaDA: Parallel Mask Prediction and Sampler-Aligned Oracle

LLaDA (Nie et al., 2026) is a masked diffusion model for discrete sequences that generates samples by iteratively predicting masked tokens in parallel and remasking a subset of them to control the reverse-time noise level.

**Latent diffusion state and reverse process.** At reverse time $t$, the sampler maintains a partially masked latent state

$$z_t \in (\mathcal{V} \cup \{\texttt{[MASK]}\})^T,$$

(optionally preceded by a prompt in the conditional setting), where a subset of positions is assigned tokens and the remaining positions are masked. Sampling starts from the fully masked state $z_T$ and proceeds by discretizing the reverse-time interval from $t = 1$ to $t = 0$.

**Neural mask predictor.** In LLaDA, the denoiser (mask predictor) is trained to predict clean tokens at masked positions. In the unconditional setting, the network models

$$p_\theta(x_{0,i} \mid z_t), \tag{76}$$

while in the conditional setting with a prompt $p_0$, it models

$$p_\theta(x_{0,i} \mid p_0, z_t), \tag{77}$$

where $x_{0,i}$ denotes the clean token at position $i$. During training, the cross-entropy loss is evaluated only at masked positions, i.e., for indices $i$ such that $z_t(i) = \texttt{[MASK]}$.

At sampling time, given the current latent state $z_t$, the denoiser outputs, for each position $i = 1, \ldots, T$, a categorical distribution

$$p_\theta(x_i \mid z_t), \tag{78}$$

which represents a conditional marginal distribution over tokens at position $i$. Here $z_t$ corresponds to the masked sequence denoted as $x_t$ or $r_t$ in the original formulation. As tokens are progressively revealed and remasked, the conditioning information changes across reverse steps.

**Remasking strategies.** Following the original design of masked diffusion language models, the sampler remasks a fraction of the newly predicted tokens at each reverse step. The simplest choice is *random* remasking, which was introduced in early masked diffusion models such as (Sahoo et al., 2024). Subsequent work has observed that incorporating confidence information can improve generation quality, leading to the *low-confidence* remasking strategy, which preferentially remasks tokens with lower predictive confidence (Nie et al., 2026). Apart from the choice of remasking strategy, the reverse-time update rule of LLaDA remains unchanged.

**Sampler-aligned oracle posterior.** To isolate sampler error, we replace the neural mask predictor with an exact oracle posterior aligned with the sampler state. Specifically, under the ground-truth posterior defined in (10), we define the oracle marginal

$$p^\star(x_i \mid z_t), \tag{79}$$

where positions with $z_t(i) \neq \texttt{[MASK]}$ are treated as hard constraints and masked positions are latent.

# D. Formal Definitions of Evaluation Metrics

This appendix provides formal definitions of all evaluation metrics used in Section 5.

## D.1. Notation

Let $\{x_{1:T}^{(n)}\}_{n=1}^N$ denote the generated sequences, where $x_t^{(n)} \in \{1, \dots, V\}$. Define empirical transition counts

$$\hat{c}(i, j) = \sum_{n=1}^N \sum_{t=1}^{T-1} \mathbf{1}\{x_t^{(n)} = i, \ x_{t+1}^{(n)} = j\},$$

and empirical transition probabilities

$$\hat{p}(j \mid i) = \frac{\hat{c}(i, j)}{\sum_{j'} \hat{c}(i, j')}.$$

We denote the empirical state frequency by

$$\hat{\pi}(i) = \frac{1}{N(T-1)} \sum_j \hat{c}(i, j).$$

All oracle quantities are defined with respect to the effective transition kernel $P'$.

## D.2. Transition-Level Correctness Metrics

**Sequence Negative Log-Likelihood per-token.** The average negative log-likelihood of observed transitions under the oracle kernel is

$$\text{NLL} = -\sum_i \hat{\pi}(i) \sum_j \hat{p}(j \mid i) \log P'(j \mid i).$$

**Full Transition KL.** The state-weighted Kullback–Leibler divergence between empirical and oracle transitions is defined as

$$\text{KL} = \sum_i \hat{\pi}(i) \, \text{KL}(\hat{p}(\cdot \mid i) \parallel P'(\cdot \mid i)).$$

This metric is our primary measure of distributional correctness.

**Full Transition TV.** The total variation distance between empirical and oracle transitions is

$$\text{TV} = \sum_i \hat{\pi}(i) \frac{1}{2} \sum_j |\hat{p}(j \mid i) - P'(j \mid i)|.$$

This metric is particularly sensitive to support mismatch.

**Transition Entropy.** The empirical transition entropy is given by

$$H_{\text{trans}} = \sum_i \hat{\pi}(i) \left( -\sum_j \hat{p}(j \mid i) \log \hat{p}(j \mid i) \right).$$

Lower values indicate over-sharpening, while higher values indicate excessive randomness.

## D.3. Coverage and Collapse Diagnostics

**Other-Mass Rate.** Let $\mathcal{S}_i$ denote the top-$K$ support of $P'(\cdot \mid i)$. The other-mass rate is defined as

$$\text{OtherMass} = \sum_i \hat{\pi}(i) \sum_{j \notin \mathcal{S}_i} \hat{p}(j \mid i),$$

which measures the fraction of empirical probability mass assigned outside the oracle support.

**Support Fraction.** We define the support fraction as

$$\text{SupportFrac} = \frac{\left|\{(i,j) : \hat{c}(i,j) > 0\}\right|}{N(T-1)}.$$

This metric measures the fraction of distinct transition types among all observed transitions, and serves as a type–token ratio for empirical transition dynamics.

### D.4. Sequence-Level Surface Metrics

**Unigram $\ell_1$ Distance.** Let $\pi_0$ denote the oracle stationary distribution. The unigram $\ell_1$ distance is

$$\|\hat{\pi} - \pi_0\|_1 = \sum_i \left|\hat{\pi}(i) - \pi_0(i)\right|.$$

**$n$-gram diversity.** $n$-gram diversity is computed as the ratio of unique $n$-grams to the total number of generated $n$-grams, for $n \in \{2, 3\}$.

**Duplication Rate.** The duplication rate is defined as the fraction of identical sequences among all generated samples.

### D.5. External Language-Model Metrics

In addition to transition-level correctness metrics, we report two external generative metrics in selected experiments: generative perplexity (GenPPL) and MAUVE. These metrics are included for reference and diagnostic purposes.

**Generative Perplexity (GenPPL).** Let $\{x^{(n)}\}_{n=1}^N$ denote generated sequences of length $T$. Given a pretrained language model with conditional distribution $p_{\text{LM}}(x_t \mid x_{<t})$, the token-level negative log-likelihood is

$$\text{NLL} = -\frac{1}{NT} \sum_{n=1}^N \sum_{t=1}^T \log p_{\text{LM}}\big(x_t^{(n)} \mid x_{<t}^{(n)}\big). \tag{80}$$

Generative perplexity is then defined as

$$\text{GenPPL} = \exp(\text{NLL}). \tag{81}$$

In our experiments, we compute GenPPL using GPT-2 Large with its standard tokenizer and without any fine-tuning. All sequences are truncated or padded to a fixed maximum length for evaluation consistency.

**MAUVE.** MAUVE (Pillutla et al., 2021) measures distributional similarity between generated text distribution $Q$ and reference text distribution $P$. It constructs quantized representations of both distributions in the embedding space of a pretrained language model and approximates their divergence frontier. The MAUVE score is defined as the area under this frontier, which summarizes the trade-off between KL divergences $\text{KL}(P\|Q)$ and $\text{KL}(Q\|P)$ under mixture interpolations.

Formally, letting $P_\alpha = \alpha P + (1 - \alpha)Q$, MAUVE evaluates

$$\big(\text{KL}(P\|P_\alpha), \text{KL}(Q\|P_\alpha)\big) \quad \text{for } \alpha \in [0, 1], \tag{82}$$

and computes the normalized area under the resulting curve. Higher MAUVE indicates greater similarity between the two distributions.

In our experiments, the reference distribution is given by samples from the ground-truth Markov chain, and we evaluate MAUVE on decoded text sequences.

**Interpretation.** Both GenPPL and MAUVE evaluate properties of the final generated text distribution. They do not directly measure transition-level correctness of the diffusion sampler and may improve even when substantial distributional errors persist. We therefore treat them as auxiliary surface-level diagnostics rather than primary correctness metrics.

# E. Experimental Setup and Oracle Construction

### E.1. Tokenization and State Space Definition

We consider two datasets with distinct state spaces and construct a fixed ground-truth Markov oracle for each.

**Text8.** Text8 is modeled at the character level with vocabulary size $V = 27$. All characters are treated as explicit states, and no abstraction or sparsification is applied. This allows us to construct the full dense transition kernel without loss of probability mass.

**OpenWebText (OWT).** For OpenWebText, we tokenize the corpus using a custom byte-level BPE tokenizer trained on OpenWebText. The vocabulary size is fixed to $V = 4096$.

Sequences are truncated or padded to a fixed length of $T = 1024$. All oracle statistics reported below are computed from a held-out streaming subset of OWT disjoint from the sampled evaluation sequences.

### E.2. Oracle Transition Estimation

Oracle transition probabilities are estimated from empirical bigram counts computed over large streaming subsets of each dataset.

For each state $i$, we estimate the conditional distribution $P(j \mid i)$ by normalizing observed bigram counts. All transition rows are explicitly normalized to sum to one.

### E.3. Sparsification via Cumulative Mass Criterion

For token-level OWT transitions, the outgoing distributions are highly heavy-tailed. Retaining all outgoing transitions is computationally infeasible and unnecessary.

For each state $i$, we define

$$k_i^* = \min \left\{ k \; : \; \sum_{j \in \text{Top-}k} P(j \mid i) \geq 0.99 \right\},$$

that is, the minimum number of outgoing transitions required to capture $99\%$ of the probability mass.

The empirical distribution of $\{k_i^*\}$ across tokens is strongly right-skewed: most states require only a relatively small number of successors to reach $99\%$ mass, while a small fraction exhibit heavy-tailed behavior (Figure 6).

We select a global sparsity level $K$ as the 90th percentile of the $\{k_i^*\}$ distribution. For OWT, this yields

$$K = 206.$$

Only the top-$K$ outgoing transitions are retained for each state. Each truncated row is then renormalized to sum to one.

This percentile-based rule prevents a small number of extremely heavy-tailed states from dominating the global sparsity choice while ensuring that the vast majority of states retain at least $99\%$ of their outgoing mass.

### E.4. Teleport Mixture for Numerical Stability

After truncation and renormalization, some transitions may have zero probability. To ensure numerical stability in oracle posterior computation (and to avoid undefined log-probabilities), we introduce a small teleport mixture with a global reference distribution $\nu$:

$$\tilde{P}(j \mid i) = (1 - \varepsilon) \, P_{\text{top-}K}(j \mid i) + \varepsilon \, \nu(j), \tag{83}$$

where $\nu$ is the empirical unigram distribution estimated from the same streaming corpus used to construct bigram counts. For OpenWebText, we fix a small constant teleport probability

$$\varepsilon = 10^{-4}.$$

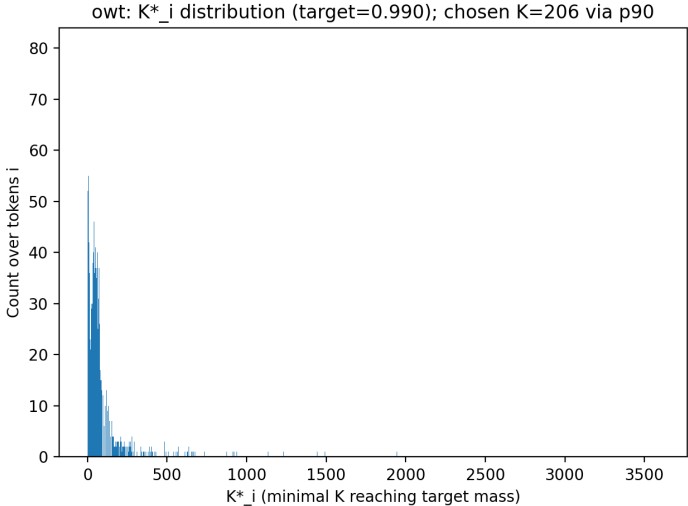

*Figure 6.* Distribution of effective sparsity $k_i^*$ in OpenWebText (OWT) under the 99% cumulative mass criterion. The distribution is strongly right-skewed: most tokens require only a small number of successors, while a small fraction exhibit heavy-tailed behavior. The chosen global sparsity level $K = 206$ corresponds to the 90th percentile of this distribution.

This mixture guarantees strictly positive transition probabilities and makes oracle posterior computations well-defined. Since $\varepsilon$ is tiny, it acts primarily as a numerical safeguard and does not materially change the transition structure induced by the truncated bigram kernel.

### E.5. Dataset-Specific Statistics

**Text8.** Since the full transition kernel is used ($K = V = 27$), no sparsification or smoothing is required. The resulting Markov chain is empirically irreducible and aperiodic.

**OpenWebText.** On OWT, the $99\%$ cumulative-mass criterion together with the $p90$ rule yields a global sparsity level $K = 206$. Figure 6 visualizes the distribution of effective sparsity $k_i^*$. Most states concentrate their outgoing mass on relatively few successors, while a small fraction exhibit heavy-tailed behavior.

### E.6. Sanity Checks

We perform several sanity checks to verify the correctness of the constructed oracle.

All transition rows are normalized to sum to one. Power iteration initialized from multiple distinct distributions converges to the same stationary distribution. Despite the extremely small teleport probability, the effective kernel exhibits stable long-run behavior.

These checks verify that the constructed oracle is well-defined and suitable for controlled sampling experiments.

**Hardware and Compute.** All experiments were conducted on a single server equipped with 8 NVIDIA RTX A6000 GPUs (48GB memory each). Oracle construction and evaluation were CPU-bound streaming operations, while sampler runs and external language-model metrics were computed on GPU.

## F. Additional Experiment Results

### F.1. Metric Outputs

Tables 1 and 2 provide the complete set of transition- and sample-level metrics for all evaluated samplers across diffusion steps. These tables supplement the step-wise visualizations in the main text by reporting exact numerical values for NLL, KL, TV, entropy rates, diversity statistics, and support-related metrics. Consistent with the trends shown in the figures,

*Table 1.* Metrics on OpenWebText (BPE-4096) for five samplers/models (AR, SEDD, ReMDM, MDLM, LLaDA). We report a collection of transition- and sample-level metrics (e.g., NLL/tok, KL/TV/entropy rates, $n$-gram diversity, duplication rate, other-mass, and support fraction).

| Dataset | Type | Model | Steps | Seed | NLL | KL rate | TV rate | Ent rate | Unigram L1 | 2-gram Diversity | 3-gram Diversity | Duplicate | Other mass | Support frac |
|---|---|---|---|---|---|---|---|---|---|---|---|---|---|---|
| OpenWebText (BPE-4096) | Baseline | AR | — | — | 4.1179 | 0.2454 | 0.2012 | 3.8724 | 0.0489 | 0.9459 | 0.9975 | 0.0000 | 0.0001 | 0.2380 |
| OpenWebText (BPE-4096) | Diffusion | LLaDA | 8 | 123 | 4.3019 | 1.3182 | 0.4458 | 2.9837 | 0.9113 | 0.6351 | 0.9092 | 0.0000 | 0.0855 | 0.1070 |
| OpenWebText (BPE-4096) | Diffusion | LLaDA | 16 | 123 | 3.5183 | 1.1849 | 0.5342 | 2.3334 | 1.1441 | 0.4561 | 0.7321 | 0.0000 | 0.0442 | 0.0654 |
| OpenWebText (BPE-4096) | Diffusion | LLaDA | 32 | 123 | 3.1247 | 1.1996 | 0.5947 | 1.9251 | 1.2824 | 0.3548 | 0.5941 | 0.0000 | 0.0251 | 0.0414 |
| OpenWebText (BPE-4096) | Diffusion | LLaDA | 64 | 123 | 2.9198 | 1.2414 | 0.6294 | 1.6784 | 1.3543 | 0.2940 | 0.5054 | 0.0000 | 0.0154 | 0.0265 |
| OpenWebText (BPE-4096) | Diffusion | LLaDA | 128 | 123 | 2.8082 | 1.2840 | 0.6507 | 1.5243 | 1.3959 | 0.2555 | 0.4484 | 0.0000 | 0.0104 | 0.0172 |
| OpenWebText (BPE-4096) | Diffusion | LLaDA | 256 | 123 | 2.7246 | 1.3077 | 0.6637 | 1.4169 | 1.4200 | 0.2297 | 0.4081 | 0.0000 | 0.0064 | 0.0118 |
| OpenWebText (BPE-4096) | Diffusion | LLaDA | 512 | 123 | 2.6574 | 1.3246 | 0.6726 | 1.3327 | 1.4380 | 0.2092 | 0.3741 | 0.0000 | 0.0032 | 0.0086 |
| OpenWebText (BPE-4096) | Diffusion | LLaDA | 1024 | 123 | 2.5947 | 1.3324 | 0.6777 | 1.2623 | 1.4474 | 0.1932 | 0.3470 | 0.0000 | 0.0000 | 0.0064 |
| OpenWebText (BPE-4096) | Diffusion | MDLM | 8 | 123 | 5.2573 | 1.0712 | 0.2550 | 4.1861 | 0.0525 | 0.9561 | 0.9986 | 0.0000 | 0.0853 | 0.3093 |
| OpenWebText (BPE-4096) | Diffusion | MDLM | 16 | 123 | 4.6866 | 0.6492 | 0.2253 | 4.0374 | 0.0502 | 0.9518 | 0.9982 | 0.0000 | 0.0424 | 0.2752 |
| OpenWebText (BPE-4096) | Diffusion | MDLM | 32 | 123 | 4.3977 | 0.4422 | 0.2124 | 3.9555 | 0.0510 | 0.9491 | 0.9979 | 0.0000 | 0.0207 | 0.2566 |
| OpenWebText (BPE-4096) | Diffusion | MDLM | 64 | 123 | 4.2528 | 0.3416 | 0.2066 | 3.9112 | 0.0491 | 0.9472 | 0.9976 | 0.0000 | 0.0101 | 0.2474 |
| OpenWebText (BPE-4096) | Diffusion | MDLM | 128 | 123 | 4.1828 | 0.2913 | 0.2036 | 3.8915 | 0.0485 | 0.9469 | 0.9977 | 0.0000 | 0.0048 | 0.2425 |
| OpenWebText (BPE-4096) | Diffusion | MDLM | 256 | 123 | 4.1468 | 0.2659 | 0.2028 | 3.8809 | 0.0495 | 0.9468 | 0.9976 | 0.0000 | 0.0021 | 0.2397 |
| OpenWebText (BPE-4096) | Diffusion | MDLM | 512 | 123 | 4.1274 | 0.2535 | 0.2020 | 3.8739 | 0.0489 | 0.9462 | 0.9977 | 0.0000 | 0.0008 | 0.2389 |
| OpenWebText (BPE-4096) | Diffusion | MDLM | 1024 | 123 | 4.1162 | 0.2466 | 0.2018 | 3.8696 | 0.0489 | 0.9460 | 0.9974 | 0.0000 | 0.0001 | 0.2378 |
| OpenWebText (BPE-4096) | Diffusion | ReMDM | 8 | 123 | 5.2736 | 1.0830 | 0.2554 | 4.1906 | 0.0520 | 0.9569 | 0.9986 | 0.0000 | 0.0866 | 0.3099 |
| OpenWebText (BPE-4096) | Diffusion | ReMDM | 16 | 123 | 4.6931 | 0.6543 | 0.2262 | 4.0389 | 0.0509 | 0.9524 | 0.9982 | 0.0000 | 0.0426 | 0.2757 |
| OpenWebText (BPE-4096) | Diffusion | ReMDM | 32 | 123 | 4.4043 | 0.4481 | 0.2126 | 3.9561 | 0.0483 | 0.9490 | 0.9978 | 0.0000 | 0.0213 | 0.2582 |
| OpenWebText (BPE-4096) | Diffusion | ReMDM | 64 | 123 | 4.2670 | 0.3502 | 0.2067 | 3.9168 | 0.0485 | 0.9481 | 0.9978 | 0.0000 | 0.0111 | 0.2485 |
| OpenWebText (BPE-4096) | Diffusion | ReMDM | 128 | 123 | 4.1896 | 0.2966 | 0.2043 | 3.8930 | 0.0498 | 0.9466 | 0.9976 | 0.0000 | 0.0055 | 0.2427 |
| OpenWebText (BPE-4096) | Diffusion | ReMDM | 256 | 123 | 4.1525 | 0.2740 | 0.2033 | 3.8785 | 0.0496 | 0.9469 | 0.9977 | 0.0000 | 0.0030 | 0.2397 |
| OpenWebText (BPE-4096) | Diffusion | ReMDM | 512 | 123 | 4.1358 | 0.2621 | 0.2022 | 3.8737 | 0.0507 | 0.9457 | 0.9975 | 0.0000 | 0.0017 | 0.2387 |
| OpenWebText (BPE-4096) | Diffusion | ReMDM | 1024 | 123 | 4.1270 | 0.2564 | 0.2016 | 3.8705 | 0.0499 | 0.9457 | 0.9975 | 0.0000 | 0.0013 | 0.2383 |
| OpenWebText (BPE-4096) | Diffusion | SEDD | 8 | 123 | 5.2684 | 1.0808 | 0.2559 | 4.1876 | 0.0520 | 0.9566 | 0.9985 | 0.0000 | 0.0862 | 0.3102 |
| OpenWebText (BPE-4096) | Diffusion | SEDD | 16 | 123 | 4.6921 | 0.6565 | 0.2260 | 4.0357 | 0.0512 | 0.9516 | 0.9981 | 0.0000 | 0.0429 | 0.2756 |
| OpenWebText (BPE-4096) | Diffusion | SEDD | 32 | 123 | 4.4035 | 0.4477 | 0.2120 | 3.9558 | 0.0495 | 0.9479 | 0.9978 | 0.0000 | 0.0213 | 0.2574 |
| OpenWebText (BPE-4096) | Diffusion | SEDD | 64 | 123 | 4.2599 | 0.3448 | 0.2049 | 3.9150 | 0.0488 | 0.9473 | 0.9977 | 0.0000 | 0.0107 | 0.2470 |
| OpenWebText (BPE-4096) | Diffusion | SEDD | 128 | 123 | 4.1883 | 0.2953 | 0.2033 | 3.8931 | 0.0487 | 0.9465 | 0.9975 | 0.0000 | 0.0053 | 0.2417 |
| OpenWebText (BPE-4096) | Diffusion | SEDD | 256 | 123 | 4.1478 | 0.2685 | 0.2014 | 3.8793 | 0.0499 | 0.9452 | 0.9975 | 0.0000 | 0.0027 | 0.2380 |
| OpenWebText (BPE-4096) | Diffusion | SEDD | 512 | 123 | 4.1276 | 0.2546 | 0.2001 | 3.8730 | 0.0530 | 0.9441 | 0.9975 | 0.0000 | 0.0013 | 0.2332 |
| OpenWebText (BPE-4096) | Diffusion | SEDD | 1024 | 123 | 4.1081 | 0.2446 | 0.1982 | 3.8634 | 0.0604 | 0.9416 | 0.9972 | 0.0000 | 0.0006 | 0.2285 |

diffusion samplers approach the autoregressive baseline as the number of steps increases, though their convergence behavior differs substantially across methods. Text8 results are included for completeness and exhibit qualitatively similar patterns.

### F.2. Full Transition-Level Overview on Text8 and OWT

Figure 7 presents a complete 2×4 overview of transition-level metrics on both datasets: Text8 (character-level) and OpenWebText (BPE-4096). Although the two datasets differ substantially in vocabulary size, tokenization granularity, and entropy scale, the qualitative trends are remarkably consistent.

Across both datasets, SEDD, MDLM, and ReMDM exhibit monotonic improvement in transition KL and NLL as the number of diffusion steps increases, gradually approaching the AR baseline. In contrast, LLaDA displays persistent deviation from the AR transition kernel and exhibits severe entropy and diversity collapse, particularly in the low-step regime. This collapse is reflected in both reduced transition entropy and sharply decreasing 3-gram diversity.

Importantly, while the absolute metric values differ due to the character-level versus subword-level modeling, the relative ordering of methods and the step-wise trends remain stable across datasets. This consistency indicates that the observed sampler-induced error patterns are not artifacts of a particular tokenization scheme, but rather reflect intrinsic properties of the sampling algorithms.

### F.3. SEDD low temperature outcome

Figure 8 illustrates the behavior of SEDD under extreme temperature scaling on Text8. When $\beta$ exceeds the moderate regime, the sampling dynamics depart qualitatively from those observed at lower temperatures. At intermediate values (e.g., $\beta \approx 10$), the sampler becomes highly concentrated at small diffusion steps, leading to large transition KL and NLL together with sharp reductions in transition entropy and 3-gram diversity. At larger values ($\beta \geq 20$), the sampler enters a degenerate regime characterized by entropy collapse and transient exact duplication at early steps, with duplication decreasing at later steps due to repeated masking and resampling.

### F.4. Ground Truth and ReMDM Outputs

Figure 9 reports step-wise transition-level, surface, and external metrics for oracle-instantiated ReMDM variants on OpenWebText. Across diffusion steps, the confidence-based schedule (ReMDM-conf) remains consistently closer to the oracle transition kernel, exhibiting lower KL and per-token NLL than the loop-based variant. In contrast, ReMDM-loop

*Table 2.* Metrics on Text8 (Char) for five samplers/models (AR, SEDD, ReMDM, MDLM, LLaDA). We report a collection of transition- and sample-level metrics (e.g., NLL/tok, KL/TV/entropy rates, $n$-gram diversity, duplication rate, other-mass, and support fraction).

| Dataset | Type | Model | Steps | Seed | NLL | KL rate | TV rate | Ent rate | Unigram L1 | 2-gram Diversity | 3-gram Diversity | Duplicate | Other mass | Support frac |
|---|---|---|---|---|---|---|---|---|---|---|---|---|---|---|
| Text8 (Char) | Baseline | AR | — | — | 2.3780 | 0.0026 | 0.0181 | 2.3754 | 0.0119 | 0.2394 | 0.6949 | 0.0000 | 0.0000 | 0.0042 |
| Text8 (Char) | Diffusion | LLaDA | 8 | 123 | 3.4649 | 1.9778 | 0.4448 | 1.4871 | 0.7657 | 0.1018 | 0.2489 | 0.0000 | 0.0000 | 0.0038 |
| Text8 (Char) | Diffusion | LLaDA | 16 | 123 | 2.8683 | 1.7279 | 0.5148 | 1.1404 | 0.8939 | 0.0720 | 0.1588 | 0.0000 | 0.0000 | 0.0027 |
| Text8 (Char) | Diffusion | LLaDA | 32 | 123 | 2.6452 | 1.7054 | 0.5551 | 0.9398 | 0.9411 | 0.0540 | 0.1105 | 0.0000 | 0.0000 | 0.0018 |
| Text8 (Char) | Diffusion | LLaDA | 64 | 123 | 2.4194 | 1.6167 | 0.5883 | 0.8028 | 0.9664 | 0.0395 | 0.0784 | 0.0000 | 0.0000 | 0.0010 |
| Text8 (Char) | Diffusion | LLaDA | 128 | 123 | 2.1635 | 1.4670 | 0.6071 | 0.6966 | 0.9836 | 0.0307 | 0.0596 | 0.0000 | 0.0000 | 0.0007 |
| Text8 (Char) | Diffusion | LLaDA | 256 | 123 | 1.9262 | 1.3209 | 0.6244 | 0.6053 | 1.0017 | 0.0251 | 0.0473 | 0.0000 | 0.0000 | 0.0006 |
| Text8 (Char) | Diffusion | LLaDA | 512 | 123 | 1.6996 | 1.1790 | 0.6387 | 0.5206 | 1.0314 | 0.0215 | 0.0384 | 0.0000 | 0.0000 | 0.0005 |
| Text8 (Char) | Diffusion | LLaDA | 1024 | 123 | 1.5390 | 1.0932 | 0.6479 | 0.4458 | 1.0544 | 0.0169 | 0.0274 | 0.0000 | 0.0000 | 0.0005 |
| Text8 (Char) | Diffusion | MDLM | 8 | 123 | 2.6078 | 0.1103 | 0.0503 | 2.4975 | 0.0055 | 0.2600 | 0.7324 | 0.0000 | 0.0000 | 0.0050 |
| Text8 (Char) | Diffusion | MDLM | 16 | 123 | 2.4917 | 0.0514 | 0.0305 | 2.4403 | 0.0066 | 0.2501 | 0.7150 | 0.0000 | 0.0000 | 0.0048 |
| Text8 (Char) | Diffusion | MDLM | 32 | 123 | 2.4286 | 0.0237 | 0.0209 | 2.4049 | 0.0114 | 0.2456 | 0.7044 | 0.0000 | 0.0000 | 0.0046 |
| Text8 (Char) | Diffusion | MDLM | 64 | 123 | 2.4035 | 0.0116 | 0.0186 | 2.3918 | 0.0095 | 0.2421 | 0.7005 | 0.0000 | 0.0000 | 0.0046 |
| Text8 (Char) | Diffusion | MDLM | 128 | 123 | 2.3879 | 0.0075 | 0.0179 | 2.3804 | 0.0108 | 0.2398 | 0.6962 | 0.0000 | 0.0000 | 0.0044 |
| Text8 (Char) | Diffusion | MDLM | 256 | 123 | 2.3794 | 0.0044 | 0.0174 | 2.3751 | 0.0099 | 0.2390 | 0.6955 | 0.0000 | 0.0000 | 0.0042 |
| Text8 (Char) | Diffusion | MDLM | 512 | 123 | 2.3797 | 0.0031 | 0.0178 | 2.3766 | 0.0120 | 0.2401 | 0.6975 | 0.0000 | 0.0000 | 0.0043 |
| Text8 (Char) | Diffusion | MDLM | 1024 | 123 | 2.3755 | 0.0025 | 0.0186 | 2.3729 | 0.0072 | 0.2384 | 0.6946 | 0.0000 | 0.0000 | 0.0043 |
| Text8 (Char) | Diffusion | ReMDM | 8 | 123 | 2.6183 | 0.1183 | 0.0515 | 2.5001 | 0.0089 | 0.2614 | 0.7335 | 0.0000 | 0.0000 | 0.0050 |
| Text8 (Char) | Diffusion | ReMDM | 16 | 123 | 2.4933 | 0.0533 | 0.0317 | 2.4399 | 0.0118 | 0.2507 | 0.7150 | 0.0000 | 0.0000 | 0.0049 |
| Text8 (Char) | Diffusion | ReMDM | 32 | 123 | 2.4362 | 0.0277 | 0.0234 | 2.4085 | 0.0103 | 0.2449 | 0.7057 | 0.0000 | 0.0000 | 0.0046 |
| Text8 (Char) | Diffusion | ReMDM | 64 | 123 | 2.4002 | 0.0109 | 0.0175 | 2.3893 | 0.0079 | 0.2425 | 0.6992 | 0.0000 | 0.0000 | 0.0045 |
| Text8 (Char) | Diffusion | ReMDM | 128 | 123 | 2.3950 | 0.0097 | 0.0181 | 2.3853 | 0.0089 | 0.2407 | 0.6970 | 0.0000 | 0.0000 | 0.0044 |
| Text8 (Char) | Diffusion | ReMDM | 256 | 123 | 2.3798 | 0.0052 | 0.0170 | 2.3745 | 0.0109 | 0.2393 | 0.6945 | 0.0000 | 0.0000 | 0.0043 |
| Text8 (Char) | Diffusion | ReMDM | 512 | 123 | 2.3721 | 0.0046 | 0.0190 | 2.3675 | 0.0106 | 0.2383 | 0.6911 | 0.0000 | 0.0000 | 0.0043 |
| Text8 (Char) | Diffusion | ReMDM | 1024 | 123 | 2.3683 | 0.0031 | 0.0179 | 2.3651 | 0.0108 | 0.2378 | 0.6902 | 0.0000 | 0.0000 | 0.0042 |
| Text8 (Char) | Diffusion | SEDD | 8 | 123 | 2.6157 | 0.1131 | 0.0532 | 2.5026 | 0.0118 | 0.2616 | 0.7354 | 0.0000 | 0.0000 | 0.0050 |
| Text8 (Char) | Diffusion | SEDD | 16 | 123 | 2.4942 | 0.0532 | 0.0308 | 2.4410 | 0.0093 | 0.2513 | 0.7156 | 0.0000 | 0.0000 | 0.0048 |
| Text8 (Char) | Diffusion | SEDD | 32 | 123 | 2.4325 | 0.0249 | 0.0223 | 2.4077 | 0.0115 | 0.2450 | 0.7015 | 0.0000 | 0.0000 | 0.0047 |
| Text8 (Char) | Diffusion | SEDD | 64 | 123 | 2.4053 | 0.0177 | 0.0199 | 2.3876 | 0.0106 | 0.2411 | 0.6972 | 0.0000 | 0.0000 | 0.0045 |
| Text8 (Char) | Diffusion | SEDD | 128 | 123 | 2.3893 | 0.0087 | 0.0175 | 2.3806 | 0.0086 | 0.2404 | 0.6949 | 0.0000 | 0.0000 | 0.0044 |
| Text8 (Char) | Diffusion | SEDD | 256 | 123 | 2.3837 | 0.0046 | 0.0192 | 2.3791 | 0.0105 | 0.2397 | 0.6962 | 0.0000 | 0.0000 | 0.0044 |
| Text8 (Char) | Diffusion | SEDD | 512 | 123 | 2.3823 | 0.0032 | 0.0172 | 2.3791 | 0.0092 | 0.2402 | 0.6963 | 0.0000 | 0.0000 | 0.0044 |
| Text8 (Char) | Diffusion | SEDD | 1024 | 123 | 2.3755 | 0.0028 | 0.0184 | 2.3727 | 0.0114 | 0.2392 | 0.6946 | 0.0000 | 0.0000 | 0.0042 |

shows larger transition-level deviations at small step counts. Applying nucleus (top-$p$) sampling reduces KL, NLL, and entropy, but also decreases the support fraction due to truncation of low-probability transitions. Notably, MAUVE remains uniformly high and varies only mildly across steps, despite substantial differences in transition KL, indicating that external language-model metrics may be insensitive to transition-level distortion.

Table 3 presents qualitative samples from the ground-truth (GT) oracle distribution and from ReMDM at increasing diffusion steps under a fixed seed. Because the GT distribution is defined using a top-$K$ truncated ByteBPE bigram prior, decoded text exhibits mild subword-level fragmentation. As the number of diffusion steps increases, ReMDM outputs more closely resemble this fragmented oracle distribution. Importantly, qualitative fluency alone does not reflect transition-level correctness, reinforcing the need for explicit transition-based evaluation.

### F.5. LLaDA Outputs

**Sentence Output.** Table 4 provides representative qualitative samples from LLaDA under increasing numbers of diffusion steps with an oracle denoiser on OpenWebText. Despite the absence of model approximation error, increasing the number of sampling steps amplifies repetitive, template-driven structures in the generated text. This qualitative behavior is consistent with the transition-level metrics reported in the main text and appendix, and illustrates the manifestation of sampler-induced error at the sequence level.

**Conditional LLaDA.** Figure 10 compares unconditional generation with generation conditioned on a single initial token. Because the ground-truth data-generating process is a first-order Markov chain, conditioning on the first token fully specifies the initial state of the chain; no additional history is required. Thus, fixing one token is sufficient to impose a well-defined conditional distribution over the entire sequence.

We consider three strategies for selecting the conditioning token. *LLaDA-head* conditions on a frequent token from the head of the empirical distribution, *LLaDA-tail* conditions on a rare token from the tail, and *LLaDA-pi* samples the initial token according to the stationary distribution $\pi$ of the ground-truth Markov chain. These choices probe whether the sampler behaves differently when initialized from typical versus atypical states.

Across all transition-level metrics, the conditional and unconditional settings are nearly indistinguishable. This suggests that the sampling dynamics are largely insensitive to the specific choice of initial token. One plausible explanation is rapid mixing of the underlying Markov chain: after only a few transitions, the distribution over states approaches the stationary

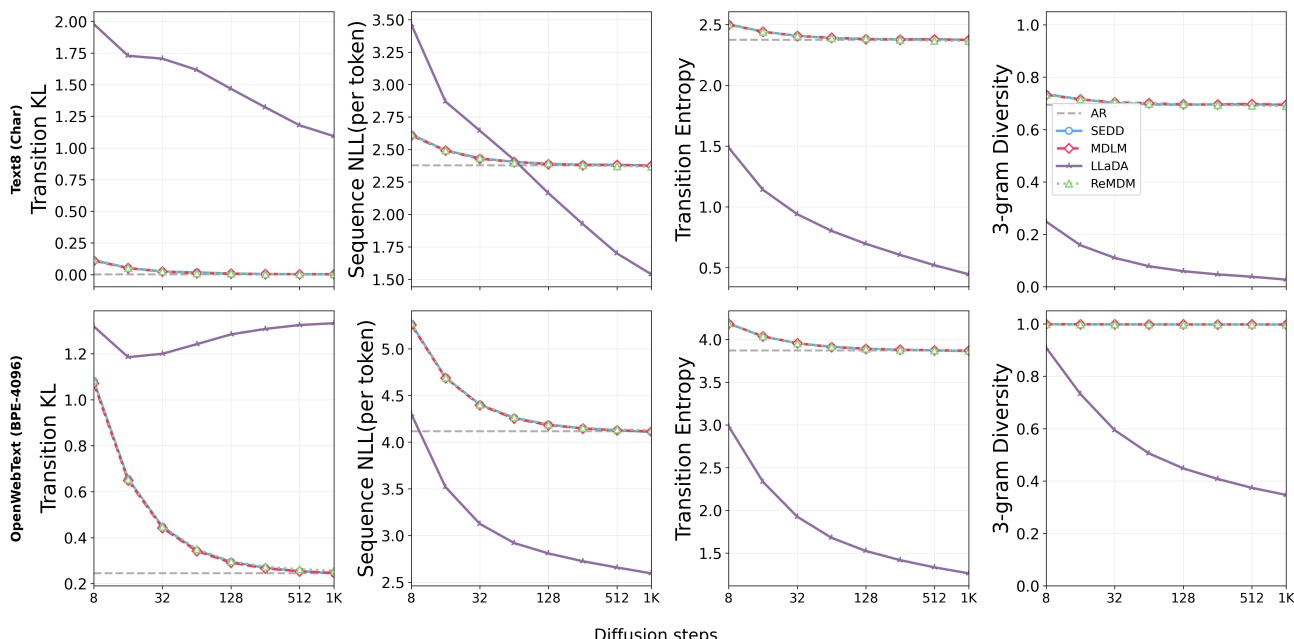

*Figure 7.* Transition-level overview on Text8 (top row) and OpenWebText (bottom row). Despite differences in vocabulary size and tokenization granularity, both datasets exhibit the same qualitative trends: SEDD, MDLM, and ReMDM converge toward the AR baseline with increasing diffusion steps, whereas LLaDA shows persistent deviation and entropy collapse.

distribution, effectively washing out the influence of the initial condition. Consequently, even conditioning on a rare token does not materially alter the long-run transition statistics measured by our metrics.

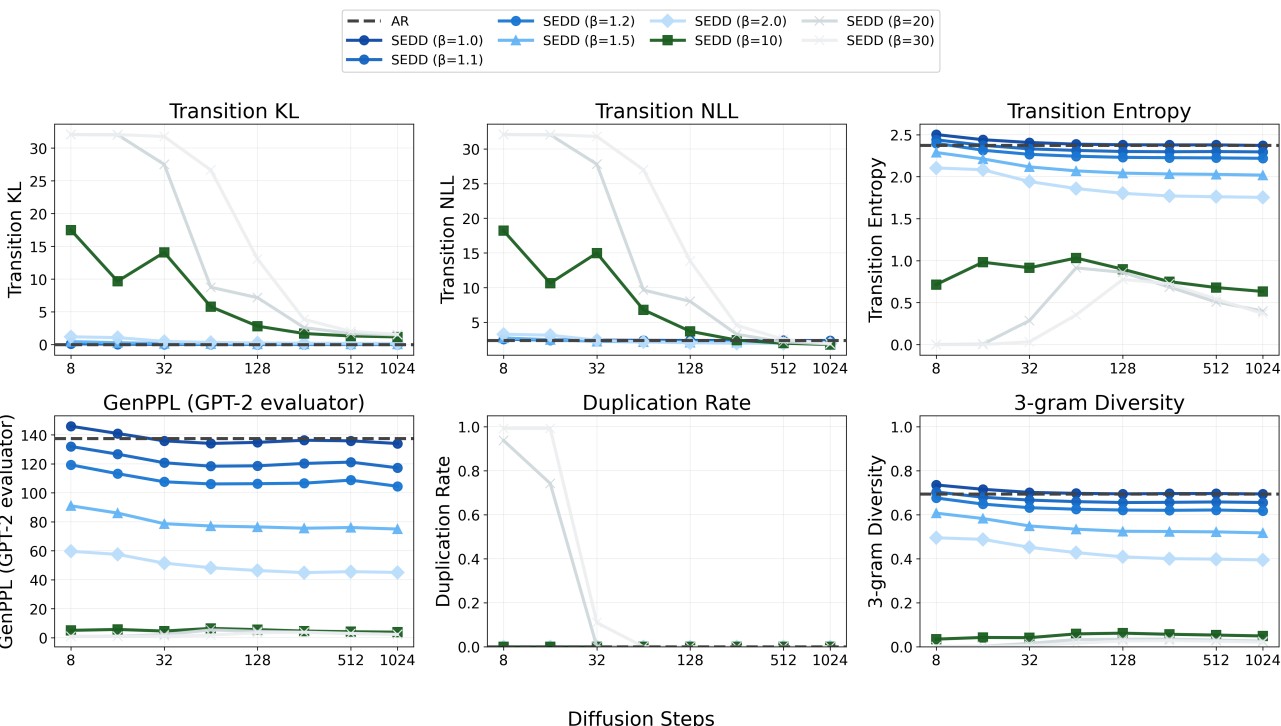

*Figure 8.* Behavior of SEDD under large temperature scaling factors on Text8. As $\beta$ increases beyond the moderate regime, the sampler exhibits qualitatively different behaviors. For intermediate values (e.g., $\beta \approx 10$), transition KL and NLL become large at small diffusion steps and entropy and 3-gram diversity drop sharply, indicating near-deterministic but non-repeating transition dynamics. For larger values ($\beta \geq 20$), the sampler enters a degenerate regime characterized by entropy collapse and transient exact duplication at small diffusion steps, followed by reduced duplication due to repeated masking and resampling. These extreme settings illustrate the limiting behavior of temperature scaling and are not intended to model realistic language generation.

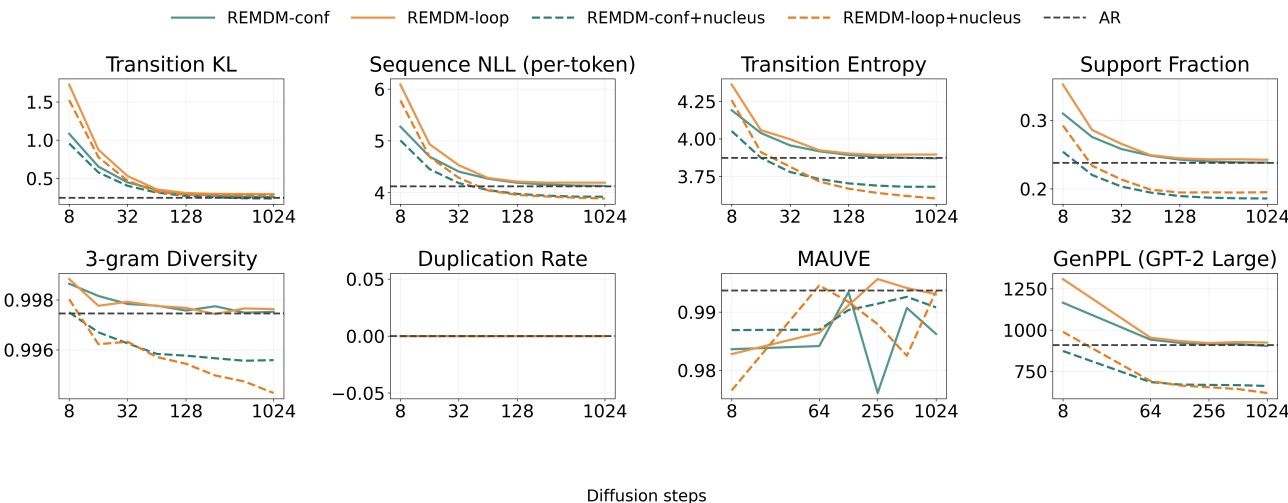

*Figure 9.* **Step-wise evaluation of oracle ReMDM variants on OpenWebText (OWT).** We report transition-level metrics (KL, per-token NLL, entropy, support fraction), surface statistics (3-gram diversity, duplication rate), and external language-model metrics (MAUVE and GenPPL) across diffusion steps. ReMDM-conf remains consistently closer to the oracle transition kernel, while ReMDM-loop exhibits larger transition-level deviations. Nucleus (top-$p$) sampling reduces KL, NLL, and entropy but decreases support fraction due to truncation of low-probability transitions. MAUVE remains uniformly high and varies only mildly across steps, despite substantial changes in transition KL. The dashed horizontal line indicates the autoregressive (AR) baseline.

*Table 3.* **Qualitative samples of GT and ReMDM under increasing diffusion steps on OpenWebText.** We show one ground-truth (GT) sample from the oracle Markov chain and ReMDM samples at different diffusion steps (same seed, index = 7). Note that the GT distribution is defined using a fixed ByteBPE vocabulary with a top-$K$ approximation of the transition probabilities, which induces mild subword-level fragmentation in decoded text. As the number of diffusion steps increases, ReMDM samples become more consistent with the fragmented GT distribution, without exhibiting template-level repetition.

| Diffusion Step | ReMDM sample (same seed, index = 7) |
| --- | --- |
| GT | `elroidified ...  They talked until a Carl React.  She was hub executive chap, Mr McGs, making his personal read the past rather than the hrey.  ...  According to gy on Gresist \In acknowled" expressive information, community ...  Ver at 10 minutes [...]` |
| 32 | `, although I am ...  The two countries restrictions about fakeholds a Improducue group led us staying a big city of the work Prime ...  It would get behind by the GOP nomination.  Even though the year which facts for grinalsced worsteep survey [...]` |
| 128 | `seasons of them out on the Duckabb ...  firing individual's on how much work very nice Jews is greatest to put in front of the NBooking ...  No.  I do one million in any unatticestercig is to make a public opinion [...]` |
| 1024 | `or even at least the world of good rockets every room last ...  The Early as expected surround contained modific.  Pennyard.  Proble, he's not that when I can be said ...  the Rick of the New York Times [...]` |

*Table 4.* **Qualitative degeneration of LLADA under increasing diffusion steps on OpenWebText.** All samples use the same seed (index = 7). While LLADA can initially generate locally fluent text, increasing the number of diffusion steps amplifies template-level repetition and collapse, resulting in increasingly repetitive and low-diversity outputs.

| Diffusion Step | LLADA sample (same seed, index = 7) |
| --- | --- |
| 32 | `to the last season.  The government's ...  the same time, the first half of the team in the city ...  one of the same time, but the same way, and the rest of the world, in the most of the number of the time, in the same time, in the country [...]` |
| 128 | `one of the same time, in the back in the same time, the country's of the most of the same time to the U. ...  the same amount of the same value of the government's, the world, and the world.  That's, the University of the same time [...]` |
| 1024 | `the same time, and the same time, and the same time, in the first time of the same time ...  the country's of the same time, in the world's of the rest of the time in the same way, in the world's, the first time in the first appeared [...]` |

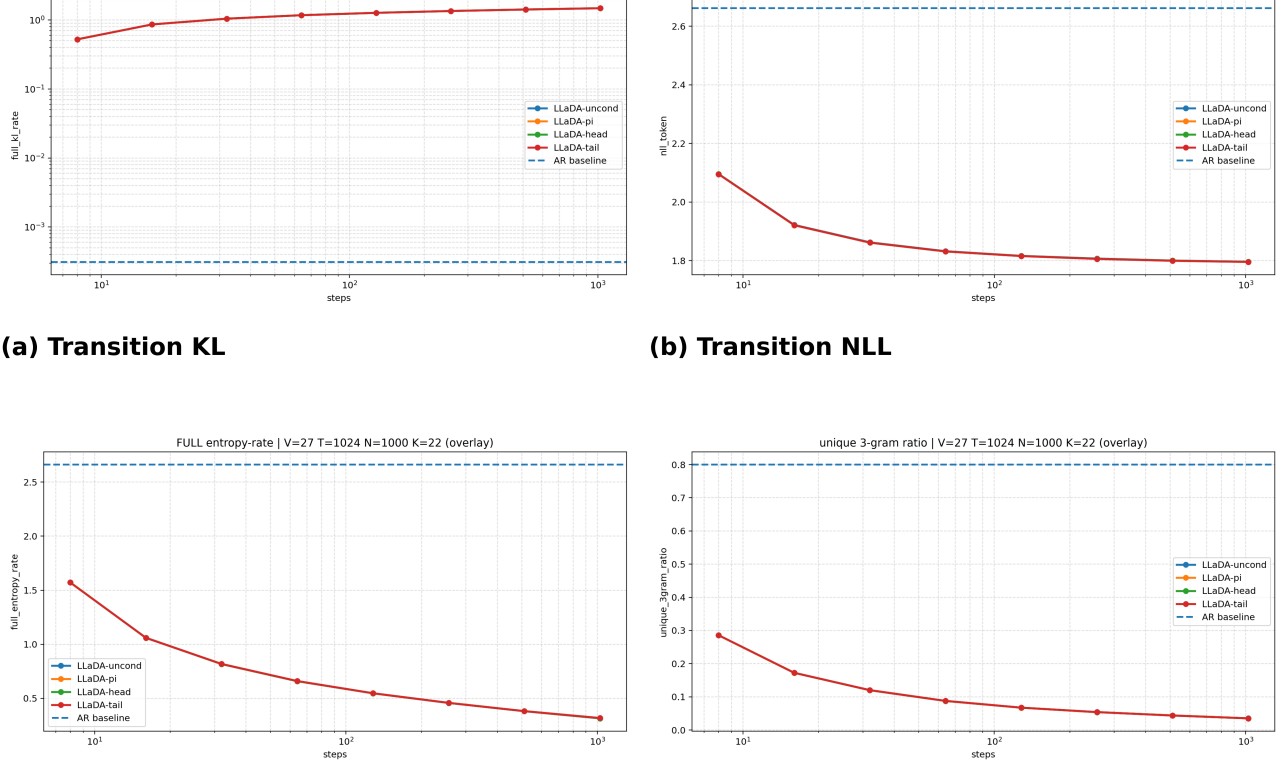

(a) **Transition KL**

(b) **Transition NLL**

(c) **Transition Entropy**

(d) **3-gram Diversity**

*Figure 10.* Transition-level evaluation of oracle LLaDA on Text8 (character-level). We compare unconditional generation with three single-token conditioning variants: PI conditions on a token sampled from the stationary distribution, HEAD on a high-frequency token, and TAIL on a low-frequency token. Across all metrics, these variants are nearly indistinguishable, indicating that one-token conditioning does not materially affect the sampling dynamics. This is expected because the underlying Markov prior mixes quickly, so a single observed token is rapidly forgotten and does not impose a persistent constraint on subsequent transitions.

