# OpenReview forum: "Is Your Diffusion Sampler Actually Correct? A Sampler-Centric Evaluation of Discrete Diffusion Language Models"
_ICML.cc/2026/Conference — ICML 2026 regular_

### Official Review · Reviewer_Fjpg · 2026-02-15

**Soundness:** 3
**Presentation:** 2
**Significance:** 3
**Originality:** 3
**Overall Recommendation:** 5
**Confidence:** 3

**Summary:**

The authors aim to distinguish decoding errors introduced by a learned diffusion language model (dLLM) from those caused by a sampler, specifically evaluating the latter. They replace learned denoisers with an oracle Hidden Markov Model (HMM) posterior derived from a ground-truth Markov chain to evaluate the sampler. They argue that most few-step samplers struggle under this evaluation, and that current metrics, such as negative log-likelihood (NLL) of the entire output and MAUVE, can be misleading.

**Compliance With Llm Reviewing Policy:**

Affirmed.

**Final Justification:**

The authors address most of my concerns through detailed discussion and analysis. While I still believe the HMM/bi-gram assumption and the underlying distribution of natural language mismatch reflect a complex issue that remains open for future work, I find the current attempt meaningful. In particular, the sampler-centric evaluation provides a novel and much-needed perspective. I appreciate this contribution and its novelty, and I raise my score from 4 to 5.

**Key Questions For Authors:**

1. Does defining ground-truth $p_0$ in a simple autoregressive manner (Equation 6) lead to bias in the evaluation of diffusion language model samplers?
2. (This is not a weakness) In Equation 14, the marginal posterior at each position is used as an oracle denoiser. While this is sufficient to test parallel samplers typically found in dLLMs (which implicitly assume conditional independence across positions), is it possible to generalize the framework to mimic joint distribution sampling for a trained sampler (e.g. [1]) in a computationally efficient manner?

**Reference**

[1] Bansal, Parikshit, and Sujay Sanghavi. "Enabling approximate joint sampling in diffusion lms." arXiv preprint arXiv:2509.22738 (2025).

**Limitations:**

1. Oracle modeling based on a simple autoregressive manner may lead to bias in diffusion language model sampler evaluation, as discussed in the first weakness.

**Strengths And Weaknesses:**

###  Strengths
1. The motivation for the problem is clear; there is an urgent need to separate these two sources of error.
2. The construction of an oracle HMM is highly suitable for this problem context.
3. Experimental results reveal interesting insights, such as the fact that metrics evaluating directly generated output (e.g., NLL and MAUVE) can be misleading. Additionally, they demonstrate that improvements in perplexity under temperature scaling do not necessarily reflect improved sampler correctness.

###  Weaknesses
1. The authors define the ground-truth data distribution $p_0$ in a simple autoregressive manner in Equation 6, which is empirically constructed using bigram counts. It is not entirely clear whether an oracle derived from this results in a bias when evaluating diffusion language model samplers. As discussed in Section 7.5, such an oracle only tests local transitions based on a single previous token, which significantly degrades the performance of the LLaDA sampler. This sampling methodology may be appropriate for the LLaDA model itself, as it originally encodes global information into its generated confidence scores.
2. Attention should be paid to several notations. For example, in line 123, $z_{1:T}$ (denoting a whole sequence) is defined as being in $\mathcal{V} \cup \{[MASK]\}$ rather than $(\mathcal{V} \cup \{[MASK]\})^T$. Similarly, in line 209, $z_t$ (representing the sequence at diffusion time $t$) is defined as being in $\mathcal{V}^T \cup \{[MASK]\}$ instead of $(\mathcal{V} \cup \{[MASK]\})^T$.
3. The experimental results presented in Section 7.4 may be misleading. The authors claim that nucleus sampling (top-$p$) helps reduce transition-level error in their oracle-based ReMDM experiments. However, the oracle construction in Section 3.1 already applies a top-$K$ criterion, which serves a similar function. If a sampler is evaluated on its ability to match a "truncated" ground truth, a sampler that also truncates will naturally perform better on distance-based metrics.

---

> ### Author Rebuttal · Authors · 2026-03-29
>
> We sincerely appreciate your insightful and professional comments. Below, we respond to your questions
> and believe our response addresses the major concerns raised.
>
> ---
>
> **W1+Q1: Possible bias from defining $p_0$ as a bigram autoregressive ground truth.**
>
> A1: We thank the reviewer for raising this question.
> This **does not introduce an oracle-induced bias** in the oracle–data relationship: ground truth sequences are sampled from the defined Markov target, and the oracle posterior for that target is exact by HMM inference. What differs across methods is how each sampler interfaces with the oracle signal.
>
> LLaDA is model-specific in this respect. As discussed in Section 7.5, unlike SEDD, ReMDM, and MDLM, its sampling mechanism uses denoiser-derived confidence scores to decide which masked positions are revealed at each step. Replacing the learned denoiser therefore changes not only token proposals, but also the confidence signal that drives reveal-order dynamics. The stronger LLaDA degradation is thus not evidence of a generic bias of the framework, but of especially strong sampler–denoiser coupling in that method.
>
> To be clear, what we call the “LLaDA” sampler is the LLaDA sampling algorithm run with the oracle denoiser for our data-generating process, not the original denoiser trained on natural language.  We will revise to make this distinction explicit.
>
> ---
>
> **W2: Attention should be paid to several notations.**
>
> A2: Thanks for the catch. We will correct these notation typos throughout the paper and carefully check for consistency in the camera-ready version.
>
> ---
>
> **W3: The experimental results presented in Section 7.4 may be misleading.**
>
> A3: Thank you for the comment, and we apologize for any lack of clarity in our presentation. The concern appears to **conflate** two distinct operations that play fundamentally different roles in our setup, and we will revise Section 7.4 to make that distinction explicit.
>
> 1. The top-$K$ operation in Section 3.1 is part of the oracle definition itself:
>
> $$
> P'(v \mid v') = (1-\varepsilon) P_{\text{top-}K}(v \mid v') + \varepsilon \nu(v).
> $$
> It is introduced only to define a sparse, tractable oracle transition kernel on OWT so that exact posterior inference is computationally feasible.
>
> 2. By contrast, the nucleus top-$p$ operation in Section 7.4 is a sampler-side heuristic from the original ReMDM paper [2], which we evaluate as part of the original ReMDM setting. These two operations therefore should not be interpreted as the same role.
>
> Accordingly, our point in Section 7.4 is **not** that truncation in general improves correctness. Rather, the point is that **MAUVE and transition-level correctness can disagree.**
>
> In the original ReMDM paper, the authors show that nucleus top-$p$ improves MAUVE, and ReMDM-loop can appear stronger than ReMDM-conf under MAUVE at sufficiently large step counts.
>
> In our oracle evaluation, adding the same original nucleus top-$p$ heuristic can lower transition KL, but ReMDM-loop still remains worse than ReMDM-conf under transition KL/NLL.
>
> This mismatch is exactly the point: MAUVE is relatively insensitive to transition-level sampler error.
>
> **New Experiment:**
> We added a new MAUVE experiment under our oracle setting here: (https://anonymous.4open.science/r/figure2-6C6D/README.md). It makes this distinction clearer: ReMDM-conf is consistently closer to the oracle kernel than ReMDM-loop under transition KL/NLL, while MAUVE remains near-saturated across variants. This is exactly the point: MAUVE and transition-level correctness can disagree.
>
> [2] Wang, G., Schiff, Y., Sahoo, S. S., and Kuleshov, V. Remasking discrete diffusion models with inference-time scaling. arXiv preprint arXiv:2503.00307, 2025.
>
> ---
>
> **Q2: Is it possible to generalize the framework to mimic joint distribution sampling for a trained sampler (e.g. [1]) in a computationally efficient manner?**
>
> A4: We thank the reviewer for this thoughtful question and for pointing us to this relevant reference. [1] explicitly distinguishes per-position marginals from true joint sampling, which is highly relevant to our paper and consistent with our sampler-centric view; we will cite it as related work.
> Our current Eq. 14 uses the exact positionwise marginal posterior as the oracle denoiser, and is therefore naturally matched to parallel samplers driven by per-position denoiser outputs. Extending the framework toward joint-style sampling is an interesting future direction. Our current view is that this would likely require additional sampler-side modeling beyond simple oracle replacement, since later updates in an ADJUST-like mechanism may no longer correspond directly to the original exact oracle posterior. We also suspect that such an extension could be substantially more expensive computationally in our setting, and we view this as an interesting direction for future work.

---

> > ### Author Rebuttal · Reviewer_Fjpg · 2026-04-01
> >
> > Thanks for the authors’ effort and detailed clarifications. In particular, I am convinced that the bigram/HMM-based oracle is internally consistent. However, I **still believe there remains a gap between this modeling assumption and realistic natural language**, which limits how strongly the conclusions transfer in practice.
> >
> > That said, I do want to emphasize that I really appreciate the authors’ sampler-centric evaluation perspective—this is a much-needed direction for better understanding dLLMs, and I find this contribution valuable.
> >
> > Given these improvements and the novelty of the evaluation framework, I am raising my score to 5.

---

> > > ### Author Response · Authors · 2026-04-02
> > >
> > > Dear Reviewer Fjpg,
> > >
> > > Thanks very much for the constructive feedback and, of course, for raising the score! We will add all suggested edits to the final version. We sincerely appreciate your thoughtful comments and valuable input that helped strengthen our paper.
> > >
> > > We **agree** that there remains a real gap between our bigram/HMM-based oracle and realistic natural language. At the same time, we hope it is helpful to clarify why we believe the result **still matters in practice**. Although natural language is far richer than a first-order Markov model, lower-order statistics such as transition frequencies are still **necessary** components of the full distribution. Our results show substantial transition-level distributional mismatch in the few-step regime under the oracle setup, indicating that these samplers fail to match this lower-order structure. Since this lower-order structure is a necessary part of the full distribution, errors at this level are informative about broader distributional mismatch.
> > >
> > > Thank you again for your careful reading and for recognizing the value of the sampler-centric evaluation perspective!
> > >
> > > Best,
> > >
> > > Authors

---

### Official Review · Reviewer_LoT7 · 2026-03-09

**Soundness:** 2
**Presentation:** 3
**Significance:** 3
**Originality:** 2
**Overall Recommendation:** 2
**Confidence:** 4

**Summary:**

This paper proposes a sampler-centric evaluation framework for discrete diffusion language models by replacing learned denoisers with an exact oracle posterior derived from a Markov chain. Using this setup, the authors argue that few-step diffusion samplers remain distributionally incorrect even under an oracle denoiser, and further show that commonly used metrics such as NLL, GenPPL, and MAUVE do not reliably reflect sampler correctness. Overall, the paper offers a novel diagnostic perspective on dLLM evaluation, especially by separating sampling error from model error.

**Compliance With Llm Reviewing Policy:**

Affirmed.

**Final Justification:**

I would like to recommend reject on this paper.
While the manuscript is tackling interesting view by decoupling sampler error and model error.
However, eventhough the model is optimal, error caused by step reduction is trivial (even in analytically).
This is a well known fact in community. However, the paper introduces the error as surprising (but not actually for me).

Also, their analysis is primarily based on 1st-order markov, which I feel primarily weak setup that has high limitation on real-world generalization.
Therefore, it is unlikely that the proposed metrics from authors will be used to analyze real samplers in the community.
However, the concept or tackling topic is highly valuable. I think the manuscript should be more powerful if it includes more real-world experiments. But at this time, it seems weak to be accepted as a good conference paper.

**Key Questions For Authors:**

1. If the goal is to evaluate samplers under a first-order Markov oracle, why not include experiments on data generated from a true synthetic first-order Markov process, rather than imposing a Markov approximation on natural language corpora?

2. Since distributional correctness is measured mainly at the transition level, to what extent do these metrics reflect failures in realistic language generation with higher-order dependencies and long-range structure? How can authors defend their simplified assumption to real-world setting?

3. The explanation for LLaDA’s strong real-language performance seems to rely on coupling between the learned denoiser and linguistic priors. Is there any direct ablation or causal evidence supporting this interpretation beyond the oracle-substitution result? This is crucial to determine whether the claims of the paper is overstated or not.

4. Please carefully address my concerns in the weakness section.

**Limitations:**

The main limitation of the paper is that its conclusions are drawn from a simplified oracle built under a first-order Markov assumption, which may not adequately capture realistic language distributions. As a result, the reported transition-level mismatch may reflect the limitation of the evaluation setup itself rather than a practically meaningful failure of the samplers in real language settings. In addition, some of the paper’s explanations for strong empirical performance in language tasks appear interpretive rather than directly validated.

**Strengths And Weaknesses:**

## Strengths

**1. Novel problem formulation**

    A key strength of the paper is its perspective of explicitly separating sampling error from model error in diffusion language models. This is a novel and useful framing, as prior evaluations based only on final generated samples can conflate these two sources of error and make sampler behavior difficult to diagnose clearly.

**2. Insightful critique of current evaluation metrics**

    The critique of current metrics such as GenPPL and MAUVE is convincing and highlights meaningful limitations of existing evaluation practices.


## Weakness

**1. Assuming 1st order Markov in “language” setup might be the problem, rather than the sampler itself**

    A central concern is that the negative results may be driven more by the first-order Markov assumption in the language setup than by the samplers themselves.

    The paper does not evaluate samplers against true language distributions, but against an oracle built by approximating Text8 and OpenWebText with first-order Markov dynamics. This is already a **very strong simplification**, and the oracle is further engineered through procedures such as top-k sparsification and teleport smoothing. As a result, the study measures correctness with respect to a constructed low-order approximation, not realistic language generation.

    This makes the conclusions much less convincing as evidence about sampler correctness in natural language, where high-order dependencies and long-range structure are essential. Therefore, I find it risky to extend the conclusions from this simplified setup to broader claims about diffusion samplers in real language settings.

**2. Speculative explanation for real-world language task performance**

    I am not convinced by the claim that samplers perform well in realistic language settings because the **''learned model corrects their errors.”** A more plausible explanation is that the denoiser is trained for the true language distribution, whereas the paper replaces it with an oracle defined under a simplified first-order Markov assumption.

    If so, the observed degradation may reflect the mismatch introduced by the experimental setup, rather than evidence that the sampler fundamentally relies on model-side correction. As presented, this explanation appears interpretive rather than firmly established, which is hard to generalize.

**3. Narrow notion of distributional correctness**

    The paper’s motivation for separating sampling error from model error is reasonable, but I am not fully convinced by its notion of distributional correctness. In practice, samplers are usually compared under the same model, and their quality is judged by the distribution of many generated samples. If a sampler consistently produces high-quality outputs, it is difficult to conclude that it is still severely mismatched to the target distribution in a practically meaningful sense.

    My main concern is that the paper reduces distributional correctness to transition-level agreement with an oracle built under a first-order Markov assumption. This makes the analysis extremely narrow.  A mismatch in transition KL under such an oracle does not necessarily imply a serious mismatch for realistic language generation, where higher-order dependencies, long-range structure, and semantic coherence are central. As a result, the paper may overstate the practical significance of its negative results: it demonstrates disagreement with a simplified oracle, not necessarily failure on the practically relevant language distribution.

## Overall Comment

    Overall, while the paper presents a novel and interesting perspective, I find some of its central claims somewhat overstated. In particular, the use of a first-order Markov assumption in the language setting feels overly restrictive, which makes it difficult to judge how well the conclusions transfer to realistic language generation.

---

> ### Author Rebuttal · Authors · 2026-03-30
>
> Thank you for the careful feedback and thoughtful questions. We believe our response addresses the major concerns raised, and **we respectfully hope you will consider updating your evaluation accordingly**.
>
> ---
>
> **Response to W1**
>
> A1: Although natural language is far richer than a first-order Markov model, lower-order statistics such as transition frequencies are still **necessary components** of the full distribution. Our result is that these samplers already fail even at that level: they do not recover the bigram statistics induced by text. If those lower-order statistics are already incorrect, then the full distribution cannot be correct. We agree that directly certifying full distributional correctness on a complex distribution like natural language would be preferable, but there are no tractable ways to do this. This is another of our core points: the heuristic metrics that are in use are not good proxies for distributional correctness.
>
> ---
>
> **Response to Q1**
>
> A2: Thanks for the suggestion. We first note that our existing setup is already self-consistent: GT sequences are sampled from the defined Markov chain, and the oracle is exact for those sequences. **There is no approximation in the oracle-data relationship.**
>
> **New Experiment**:
> We run the full pipeline on a purely **synthetic** Markov chain (randomly generated transition matrix, $V=32, K=16, T=128, N=512$, averaged over 3 seeds) to provide a complementary sanity check: (https://anonymous.4open.science/r/table1-5E63/README.md)
>
> The results are fully consistent with our Text8/OWT findings: AR≈GT, all samplers show large mismatch at few steps, and by steps (=$T$), MDLM, SEDD, and ReMDM-conf largely converge toward GT, whereas LLaDA remains clearly mismatched. This confirms that the conclusions are not specific to text-derived transition statistics.
>
> Our original choice to use text8 and OWT was deliberate: we wanted targets that retain realistic vocabulary size, frequency skew, and transition statistics, while still permitting exact oracle inference. The synthetic experiment complements this by showing the same patterns hold for arbitrary transition structures.
>
> ---
>
> **Response to W2**
>
> A3: Thanks for the question. We **do not** make the claim "samplers perform well in realistic language settings because a learned model generally *corrects* their errors".  Rather, for LLaDA our intended interpretation is: the sampling mechanism uses denoiser-derived confidence scores to decide which masked positions are revealed at each step. Thus, its behavior depends not only on the predicted token distribution, but also on the structure of the confidence signal used for position selection. The original learned LLaDA denoiser, trained on natural language, may provide a much richer confidence landscape than our first-order oracle posterior.
>
> In this sense, we interpret the distinct LLaDA trend in Figure 1 not as evidence that learned models generally “correct” sampler errors, but as evidence that the LLaDA sampling mechanism is especially sensitive to the confidence signal provided by the denoiser, i.e., that its sampler–denoiser coupling is particularly strong and intrinsic to the LLaDA sampler design. We will revise any potentially ambiguous wording in the paper for clarity.
>
> ---
>
> **Response to Q3**
>
> A4: As clarified in A3, our claim is about strong sampler–denoiser coupling in LLaDA, not a generic coupling between a learned denoiser and “linguistic priors.” Our oracle intervention directly tests this claim: when the denoiser interface is replaced, LLaDA’s sampling behavior changes substantially, demonstrating strong sensitivity to that interface.
>
> ---
>
> **Response to W3+Q2**
>
> A5:
> We respectfully **disagree** that consistently high-quality outputs are sufficient to dismiss substantial distributional mismatch as practically unimportant. This is precisely the premise our paper tests: output-level metrics can improve even as the sampling distribution moves farther from the target. In Section 7.3 and Figure 3, we show an experiment in a simple autoregressive bigram model: local sharpening moves samples away from the true distribution, yet GenPPL improves monotonically while 3-gram diversity and sentence entropy collapse. Thus, even outside diffusion, favorable output-level metrics need not imply faithful sampling of the underlying distribution.
>
> As noted in A1, transition-level agreement is a **necessary** condition for exact matching to our oracle target: if the generated distribution exactly matched the target distribution $P$, then its induced transition distribution would also have to match that of $P$. Thus, violating transition-level agreement already yields a genuine negative result within our framework. Even in this controlled setting, moreover, it is difficult to identify a tractable estimator that vanishes if and only if generated samples exactly match the full target distribution $P$.

---

> > ### Author Rebuttal · Reviewer_LoT7 · 2026-04-03
> >
> > I have carefully re-examined the paper and the authors' responses. While I now better understand the specific point being made, I remain skeptical as to whether this manuscript provides a sufficiently strong contribution to warrant a top-tier acceptance. My primary concern is that more extensive experiments may be required to move beyond what feels like a somewhat intuitive conclusion.
> >
> > Regarding the dataset construction, I acknowledge my previous misunderstanding. The authors did not assume language is a first order Markov process but rather used linguistic statistics to build a first order Markov dataset. **However, I still question the necessity of using language data for a first order setup if the resulting environment is not representative of language itself.** While I understand the desire to maintain a large vocabulary and a skewed distribution, treating features extracted from non-Markovian data as a first order process feels counter-intuitive and may create unnecessary friction for the reader.
> >
> > From the perspective of those researching discrete diffusion language models, the fact that sampling error exists is well known or trivial. It is expected that these errors amplify when multiple tokens are decoded simultaneously due to the marginal joint mismatch mentioned in the related works. Consequently, demonstrating that these errors persist even in a first order Markov setting does not necessarily provide a high impact revelation. It seems obvious that the error would be even more pronounced in actual natural language environments. I believe it would have been more insightful to explore environments with higher order dependencies or perhaps to create an (near) optimal denoiser by overfitting a model to specific language data to bridge the gap between this synthetic setup and real world applications.
> >
> >
> > ### First Question
> >
> > My first question concerns the derivation of the optimal denoiser in Section 3. These derivations occupy a portion of the paper, yet they appear quite standard to researchers in this field. **I would like to ask if the authors consider this mathematical derivation to be a primary contribution of the work. If so, please explain why this derivation was particularly challenging to obtain and justify why an oracle model based on such a simple first order setup would be valuable for future research. If it is not a primary contribution, the space allocated to it seems disproportionately large**.
> >
> > ### Second Question
> >
> > My second question relates to the practical applicability of these findings. **Suppose we compare Sampler A and Sampler B. If Sampler A proves superior in your oracle setup but Sampler B performs better in a real language environment according to standard metrics, how should the community interpret this result?**
> >
> > If the authors argue that Sampler A is truly better (in distributional correctness) and that existing metrics are simply failing to capture its quality, then this paper offers significant value. However, if the result implies that we cannot determine which sampler is better in real-world setup, then the paper merely confirms a somewhat trivial property of sampling erros in a toy (1st-order markov) setup. I remain curious why the authors did not choose to use an approximated oracle via overfitting on real language to provide more meaningful guidance for practical language tasks.
> >
> > ### Third Question (Added after modification / Similar to the second one)
> >
> > The authors provide a compelling critique of how current metrics (GenPPL, MAUVE, NLL) can be "gamed" via sharpening while sacrificing distributional fidelity. To make these insights actionable for practitioners working in the "real world," what specific refinements or supplementary protocols do you propose? Since we cannot compute an exact Oracle Transition KL for natural language, how should researchers bridge the gap between your diagnostic metrics and practical performance evaluation?
> >
> > Specifically, consider a "Sampler C" that achieves state-of-the-art results on your proposed metrics (low Transition KL and Entropy) in the 1st-order environment but performs poorly on standard metrics like GenPPL and MAUVE in real-world language tasks. Can the developer of such a sampler legitimately claim it is a "universally superior" sampler?
> >
> > Overall, I feel the manuscript can be strengthen further by examining more complex scenarios or suggesting practical solutions (e.g., metrics) to bridge the gap between distributional correctness and metric alignment. Decoupling the sampling error in a first-order markov setting seems a little weak, given that the existence of sampling error is trivial (even analytically) particularly when steps are reduced.
> >
> > I am willing to change my stance if the authors can address above questions thoroughly. I will also discuss with other reviewers and AC with this concern seriously.

---

> > > ### Author Response · Authors · 2026-04-03
> > >
> > > > I have carefully re-examined the paper and the authors' responses... feels like a somewhat intuitive conclusion.
> > >
> > > > From the perspective of those researching discrete diffusion language models, the fact that sampling error exists is well known or trivial.
> > >
> > > > It seems obvious that the error would be even more pronounced in actual natural language environments.
> > >
> > > Thank you for taking the time to carefully consider these questions. There's a big and somewhat nebulous question that you are tapping into here: do most discrete diffusion researchers actually understand the limitations of these approaches for sampling?
> > > - The ICML 2024 best paper award for SEDD is anchored by Fig.  1, which measures unconditional sampling performance by plotting "Generative perplexity" (Gen PPL), showing that discrete diffusion can achieve the same number as AR with a fraction of the function evaluations. I must admit that I was quite taken by this result, and thought that discrete diffusion was actually sampling with similar performance to AR. I didn't realize that Gen PPL could be trivially gamed, and I think it took a long time for that to become understood. And, even though several papers have pointed this out, I still see this used as a primary metric to give "SOTA results" for discrete diffusion, with no other metrics.
> > > - More careful recent papers, like ReMDM (NeurIPS 2025), move away from Gen PPL. But they rely on MAUVE, which is also imperfect, and should not be mistaken for a general distributional measure of sampling performance.
> > > - The LLADA abstract says "Across extensive benchmarks, LLaDA [outperforms] our self-constructed ARM baselines." The baselines where it excels are mostly task-specific, but the impression given is that LLADA is approaching the overall capability of AR models.
> > > - In the influential "block diffusion" paper, they say "Block diffusion overcomes key limitations of both approaches..." While the claims in the paper are careful and concrete, still there's a suggestion that these approaches can sample just as well as AR models.
> > >
> > > It's frustrating to only see overly positive claims, because while I think some subset of researchers understand and find it intuitive that there are deep shortcomings, especially for unconditional sampling, there are no concrete metrics or results shown to point this out. This sub-field is, for some reason, very resistant to synthetic or semi-synthetic metrics that can easily resolve these questions. So, at a high level, I agree that the metrics that we propose are straightforward, and the results are not surprising! What is surprising is that we can find no record of these shortcomings in the literature.
> > >
> > > We will try to prepare a response to your other points in the remaining time window available, but I (PI) just wanted to share this high level thought right away, as I am interested to hear the perspectives of other researchers in the field on these concerns. Perhaps there are papers flying under the radar that clearly show the "obvious" shortcomings, in which case I would like to amplify these works so that the field can recognize and address the current shortcomings of diffusion language models.
> > >
> > > **Response to additional questions:**
> > > >Q1
> > >
> > > HMM inference itself is not the main novelty. Our contribution is to use the exact HMM posterior as an oracle denoiser, replacing learned NN outputs to isolate sampler error from denoiser/model error. Since this construction is central, we included only its essential defining formulas in the main text and deferred the derivation to Appendix A.
> > >
> > > >Q2+Q3
> > >
> > > Sampler C is plausible. More generally, we would frame this in the spirit of modern LLM benchmarking: evaluation is typically multi-metric, not determined by any single number. The strongest models or samplers should perform well across all relevant axes. Our contribution is to add one such axis by verifying whether low-order moments are captured exactly, while asking that other heuristics be matched as well.
> > >
> > > Recent new work supports this view. Paper (Fang et al., arXiv:2604.00375) suggests that confidence remasking favors one-shot quality, while random remasking supports broader multi-sample exploration. Paper (Ni et al., arXiv:2601.15165) suggests that confidence-driven arbitrary-order decoding can cause “entropy degradation”: single-sample quality may remain strong, even as the reasoning space is prematurely narrowed and multi-sample scaling worsens. Our framework goes one step further by directly isolating and measuring these hidden sampler-side errors with an exact oracle.
> > >
> > > >More complex scenarios or practical metrics would strengthen the manuscript.
> > >
> > > We agree that practical extensions and more complex controlled settings are important future directions. More broadly, our methodology can serve as a practical diagnostic tool for identifying sampler-side failure modes. Another interesting, complex setting where denoisers can be characterized is random CSPs (arXiv:2603.20589).

---

### Official Review · Reviewer_4BKw · 2026-03-14

**Soundness:** 4
**Presentation:** 4
**Significance:** 4
**Originality:** 3
**Overall Recommendation:** 5
**Confidence:** 3

**Summary:**

The paper proposes an alternative to the classical discrete diffusion language models and circumvents the problem of using an ill-posed metric to measure them. This is done by instead having a Kalman filter-style sampling oracle.
This approach exhibits good results on par with SOTA.
Overall, this is a fantastic paper.

**Compliance With Llm Reviewing Policy:**

Affirmed.

**Key Questions For Authors:**

* The results in Appendix A.7 are very interesting about the numerical stability of log and can actually have a large impact on people dealing with Monte Carlo sign problems and stability. I would consider somehow incorporating it in the main text.

* While you are building the framework with HMM, can you hypothesize what would happen for non-Markovian models, i.e., where you have correlation?

* Likewise, the slip side of the above question is, how well does your methodology work in the presence of massive class imbalance data or data with heavy tails on it's distervbution

**Limitations:**

Yes

**Strengths And Weaknesses:**

The paper is novel in how it incorporates these state space models as a form to evaluate and train dLLMs, and this is shown in a very robust and sound way. Furthermore, this also allows the models to be distributionally correct. This is shown in a technically grounded way and written extremely clearly. The authors also give all the required background, which is highly appreciated. While technically, the authors also present the methodology as a coherent, overarching idea.
I think the notion of the Oracle-ReMDM-loop is extremely valuable and can truly contribute to understanding the effects of self regularization, and will even have an impact on things like weight watchers and heavy-tail semiempirical method.
The approach of combining hidden-state oracles to sample and adjust performance and training is novel. I have not seen it before, and I think it could also have a large impact on speculative decoding and some other aspects of understanding test time compute. All of the above leads to this being an original work

---

> ### Author Rebuttal · Authors · 2026-03-29
>
> We sincerely thank Reviewer 4BKw for the positive and thoughtful review. We are glad that you found the paper technically sound and potentially impactful. Below, we respond to your questions and suggestions.
>
> ---
>
> **Q1: The results in Appendix A.7 are very interesting about the numerical stability of log and can actually have a large impact on people dealing with Monte Carlo sign problems and stability. I would consider somehow incorporating it in the main text.**
>
> A1: We thank the reviewer for highlighting the broader significance of Appendix A.7. We agree that this numerical-stability aspect is important, not only for our setting but more generally for reliable finite-precision inference. In our case, the log-domain implementation is essential for stable oracle computation over long sequences, as it prevents underflow in forward–backward recursions and enables numerically stable log-sum-exp evaluation. We will revise the main text to briefly emphasize this point and direct readers to Appendix A.7 for the full discussion.
>
> ---
>
> **Q2: While you are building the framework with HMM, can you hypothesize what would happen for non-Markovian models, i.e., where you have correlation?**
>
> A2: We thank the reviewer for this thoughtful question. Our current oracle construction uses a HMM structure because it enables exact posterior inference and thus a clean isolation of sampler error. More generally, the framework does not fundamentally require Markovianity, but it does require a tractable oracle posterior.
>
> For non-Markovian models with longer-range correlations, similar ideas may still apply in structured cases, but exact oracle inference would typically become substantially more expensive or intractable. In particular, once the HMM-style chain factorization no longer holds, one can no longer use the forward-backward algorithm to compute the exact posterior efficiently. Even in our current HMM setting, inference is already nontrivial at our sequence lengths and vocabulary sizes, which partly motivates the top-$K$ construction. In more general correlated settings, approximate oracle inference would likely be necessary, but this would introduce an additional source of error beyond the sampler itself. This is precisely why we use the HMM setting here: it allows exact posterior computation and a clean isolation of sampler error.
>
> ---
>
> **Q3: Likewise, the slip side of the above question is, how well does your methodology work in the presence of massive class imbalance data or data with heavy tails on it's distervbution?**
>
> A3: Thank you for this question. Regarding heavy tails and class imbalance, this concern is distinct from the non-Markovianity issue above: heavy-tailed or highly imbalanced distributions do not by themselves invalidate the framework, as long as exact oracle inference remains tractable.
>
> OWT is already a materially non-uniform setting rather than a light-tailed one, with pronounced long-tail behavior and substantial local imbalance; we include supplementary figures with a $k_i^*$ histogram as direct evidence and a cross-dataset Text8--OWT comparison here: (https://anonymous.4open.science/r/figures-53F9/README.md). In addition, under our oracle-setting comparisons, the same broad qualitative trends appear in both Text8 and OWT despite their markedly different distributional regimes: as the number of diffusion steps increases, SEDD, MDLM, and ReMDM move progressively toward the AR baseline, whereas LLaDA remains qualitatively distinct, with persistent transition-level deviation and stronger entropy/diversity collapse. Thus, the diagnostics remain informative not only in the simpler Text8 setting, but also in the substantially more skewed OWT setting.
>
> Therefore, our answer is yes: the methodology remains applicable and informative under substantial class imbalance and heavy-tailed structure, provided exact oracle inference is still tractable.

---

> > ### Author Rebuttal · Reviewer_4BKw · 2026-04-04
> >
> > Thank the authors for the clarifications and for providing all the added details. I will keep my score and think this should be accepted.

---

> > > ### Author Response · Authors · 2026-04-04
> > >
> > > Dear Reviewer 4BKw,
> > >
> > > We sincerely thank you for your thoughtful follow-up and positive assessment. We are very glad that our clarifications helped address your questions. Your positive assessment is a great encouragement and support to us. We will incorporate the suggested edits in the final version.
> > >
> > > Best,
> > >
> > > Authors

---

### Decision · Program_Chairs · 2026-04-30

**Decision:**

Accept (regular)

**Comment:**

This paper proposes a sampler-centric evaluation framework for discrete diffusion language models (dLLMs) by replacing learned denoisers with a Hidden Markov Model (HMM) to isolate sampler-induced error from model approximation error. The reviewers agreed that decoupling these two sources of error provides an interesting perspective, highlighting the paper's critique of how metrics like Generative Perplexity and MAUVE can be "gamed" via local sharpening. While Reviewer LoT7 recommended rejection, arguing that the existence of sampler error is "obvious" and that the 1st-order Markov setup is too simple for natural language, the other reviewers found the method robust and the analysis insightful. In the rebuttal, the authors argued that transition-level agreement is necessary for ensuring correctness of the full distribution; if a sampler fails on bigram statistics, it cannot be correct for natural language. This satisfied Reviewer Fjpg, who raised their score to a 5. Given the strong support from two out of three reviewers and the importance of establishing rigorous evaluation pipelines as dLLMs gain momentum, the submission is recommended for acceptance.